# Moving in on human motor cortex. Characterizing the relationship between body parts with non-rigid population response fields

**Wouter Schellekens**[1,2]*, **Carlijn Bakker**[1], **Nick F. Ramsey**[1], **Natalia Petridou**[2]

1 Department of Neurology and Neurosurgery, Brain Center, UMC Utrecht, Utrecht, Netherlands,
2 Radiology department, Center for Image Sciences, UMC Utrecht, Utrecht, Netherlands

* W.Schellekens@umcutrecht.nl

**Data Availability Statement:** All raw data are available at link: doi:10.18112/openneuro. ds003972.v1.0.0 The non-rigid pRF and graph

## Abstract

For cortical motor activity, the relationships between different body part representations is unknown. Through reciprocal body part relationships, functionality of cortical motor areas with respect to whole body motor control can be characterized. In the current study, we investigate the relationship between body part representations within individual neuronal populations in motor cortices, following a 7 Tesla fMRI 18-body-part motor experiment in combination with our newly developed non-rigid population Response Field (pRF) model and graph theory. The non-rigid pRF metrics reveal somatotopic structures in all included motor cortices covering frontal, parietal, medial and insular cortices and that neuronal populations in primary sensorimotor cortex respond to fewer body parts than secondary motor cortices. Reciprocal body part relationships are estimated in terms of *uniqueness*, *clique-formation*, and *influence*. We report unique response profiles for the knee, a clique of body parts surrounding the ring finger, and a central role for the shoulder and wrist. These results reveal associations among body parts from the perspective of the central nervous system, while being in agreement with intuitive notions of body part usage.

## Author summary

While over half the human brain shows elevated levels of activity during motor tasks, a complete understanding of cortical motor activity is still lacking. Although somatotopic organizations of sensorimotor cortices have been demonstrated before, a somatotopy is only the 'tip of the iceberg' for cortical motor activity. Small ensembles of neurons—even in primary sensorimotor cortex—respond to movements of multiple body parts. This raises the following questions: how are movements of different body parts structured within cortical response profiles, and how do body parts relate to each other from the human brain's perspective? Here we investigate the intrinsic structure of small neuronal populations in human motor cortices following an 18-body-part motor task, using our newly developed non-rigid population Response Field (pRF) method and high-field

theory analyses are available at link: https://github.com/wschelle/NRpRF.

**Funding:** This work was supported by the National Institute Of Mental Health of the National Institutes of Health under Award Number R01MH111417 (https://grants.nih.gov/grants/funding/r01.htm), awarded to N.P. The funders had no role in study design, data collection and analysis, decision to publish, or preparation of the manuscript.

**Competing interests:** The authors have declared that no competing interests exist.

functional MRI. We report somatotopic structures in all sensorimotor cortices, and smaller response field sizes in primary sensorimotor cortex compared to secondary motor cortices. Furthermore, we show that physically adjacent body parts are often represented in each other's cortical response field. Using graph theory, we reveal relationships between cortical body part representations, such as the relatively increased influence of the shoulder and wrist within neuronal populations during any of the 18 motor conditions.

## 1. Introduction

When we move an individual limb or body part like one of our fingers, many different cortical areas in frontal and parietal lobes show elevated levels of activity [1–4]. However, it is far from clear how the many different brain regions contribute to motor output. Even in primary motor cortex (M1), which shows the highest correlation with localized muscle activity [5,6], it is not fully understood how the neuronal activity contributes to the actual movement [7–9]. Exemplary of this lack in understanding is that M1 has been reported to exhibit both a somatotopic organization (i.e. the orderly topography of cortical body part representations, [4,10–14]), as well as efferent connections exceeding the range of individual body parts or localized muscle groups [15–17]. In our previous study, we proposed that a Gaussian population Receptive Field (pRF) model may help to reconcile these multiple M1 interpretations [18]. Our pRF model showed that M1 neuronal populations (i.e. small ensembles of neurons within MR-voxels) can both contain a preferred finger representation (pRF center) constituting the somatotopy, as well as connections to adjacent fingers reflected by the pRF size. How fingers or other body parts relate to each other within small neuronal populations can illustrate how motor cortices are wired and what functions they perform with respect to individual body part movements. Since many body parts can move in conjunction, the mutual relation between different body parts is not trivial. Our previous study investigated the movement of fingers only and, additionally, assumed a rigid order of fingers (from thumb to little finger), predefining the internal pRF structure. The limited number of body parts in combination with an *a priori* assumption on their reciprocal relations prevents quantification of body part relationships. Thus, while our previous study indicates that pRF modeling is able to model cortical motor activity, it is unknown how body parts relate to each other and how body parts are ordered within the response profile of neuronal populations.

In the current study, we investigate the relationship between body part representations in human motor cortices following an 18-body-part motor task, using pRF modeling and high-field 7 Tesla Blood-Oxygenation-Level-Dependent (BOLD) fMRI. At this point we note that in light of cortical motor activity the term 'population Response Field' is better suited than 'population Receptive Field', since cortical motor activity cannot be solely receptive in nature. Hence, the abbreviation pRF will refer to 'population Response Field' from here on. We postulate that reciprocal relationship between body parts can be elucidated by estimation of the internal structure of whole-body pRFs. Conventional pRF modeling tries to fit a Gaussian pRF across a rigid functional space, e.g. visual field locations [19] or auditory frequencies [20], which has also been applied to finger space in combination with somatosensory [21,22] and motor tasks [18]. However, to adequately assess the internal structure of neuronal populations with respect to motor activity, we cannot simply assume that each pRF consists of a rigid ordering or body parts, e.g. similar to the conventional cortical homunculus ordering of body parts [23]. Therefore, we developed a novel non-rigid pRF method that does not assume a rigid ordering of body parts. Rather than fitting a variable Gaussian function along an

unchanging dimension of body parts, variably positioned body parts are fitted within a static Gaussian shaped pRF. The non-rigid pRF method can be regarded as a Gaussian shaped theoretical response field, which is populated with a set of functions (body part movements in the current study). Properties that are common to conventional pRF methods, such as pRF center and size, can likewise be extracted from the non-rigid pRF method on the basis of position, number and spread of functions within the theoretical response field. Additionally, the non-rigid pRF approach allows for the investigation of pRF composition: one can address which body parts constitute the total pRF, including the proximity between body parts. Thus, the novel non-rigid pRF center allows for estimation of conventional pRF properties such as pRF center and size, and allows for the investigation of occurrence and proximity of body part representations within a pRF without making assumptions on the intrinsic structure of the pRF.

In order to estimate a whole-body pRF, eighteen body parts were selected for movement that encompass the lower limb, midsection, upper limb and face. The distribution of selected body parts is not uniform in terms of physical size, but was instead determined by the ability to be moved on cue. Therefore, the upper limb and face consist of more body parts that are cued for movement, compared to the lower limb and midsection. In order of appearance on the cortical homunculus, those body parts are: toes, ankle, knee, abdomen, shoulder, elbow, wrist, little finger, ring finger, middle finger, index finger, thumb, forehead, eyelid, nostril, lip, jaw, and tongue (Fig 1A). Each neuronal population will represent these body part movements within its pRF in its own unique way. Through averaging the pRFs from neuronal populations with the same body part preference (i.e. pRF center), the mean body part pRF is obtained, which represents the average response profile for any given body part movement. The relationship between body parts can then be assessed with graph theory on the basis of the mean body part pRF [24–26]. Whole-body graphs are constructed by correlation of the mean body part pRFs, representing the linkage and connection strength between body parts. For each body part representation we calculate graph theory metrics that reflect relevant aspects of body part relations: the connectivity (degree), clustering coefficient and betweenness centrality coefficient. The connectivity metric estimates the connectedness of body parts based on the similarity of their respective mean body part pRFs: the larger the connectivity, the more similar a body part's response field is compared to other body parts. The clustering metric is a measure of 'clique-formation', representing the interconnectedness of a body part and its neighboring body parts [27,28]. Betweenness centrality is measure of body part influence: here it represents the (indirect) involvement of a particular body part when other body parts move [29,30]. Lastly, we define modules of body part representations, based on shared characteristics of the mean body part pRFs (Fig 1B and 1C), using Louvain modularity [31,32].

In the current study, we investigate the relationships among 18 different body parts in the following cortical areas related to motor control: primary motor cortex (M1), primary somatosensory cortex (S1), supplementary motor area (SMA), dorsal and ventral premotor cortex (PMd and PMv, respectively), insular cortex (Insula), and superior and inferior parietal cortex (sPC and iPC, respectively). Body part relationships are scrutinized in several distinct ways. Using our novel non-rigid pRF model, we first estimate pRF center and size, approximating the neuronal population's body part preference and the size of the population's response field. We hypothesize that the non-rigid pRF centers reveal somatotopic structures in cortical motor areas that have previously been reported to exhibit a somatotopy: M1, S1, SMA and the insula [4,11–14]. Additionally, we hypothesize that the non-rigid pRF sizes will be smallest for primary sensorimotor cortices (M1 and S1), since activity profiles from primary sensorimotor cortices are thought to correlate to individual body parts to a greater extent than activity from secondary motor cortices [33,34]. Second, we quantify relationships between body parts as observed within the non-rigid pRF. We hypothesize that body parts that are adjacent on the

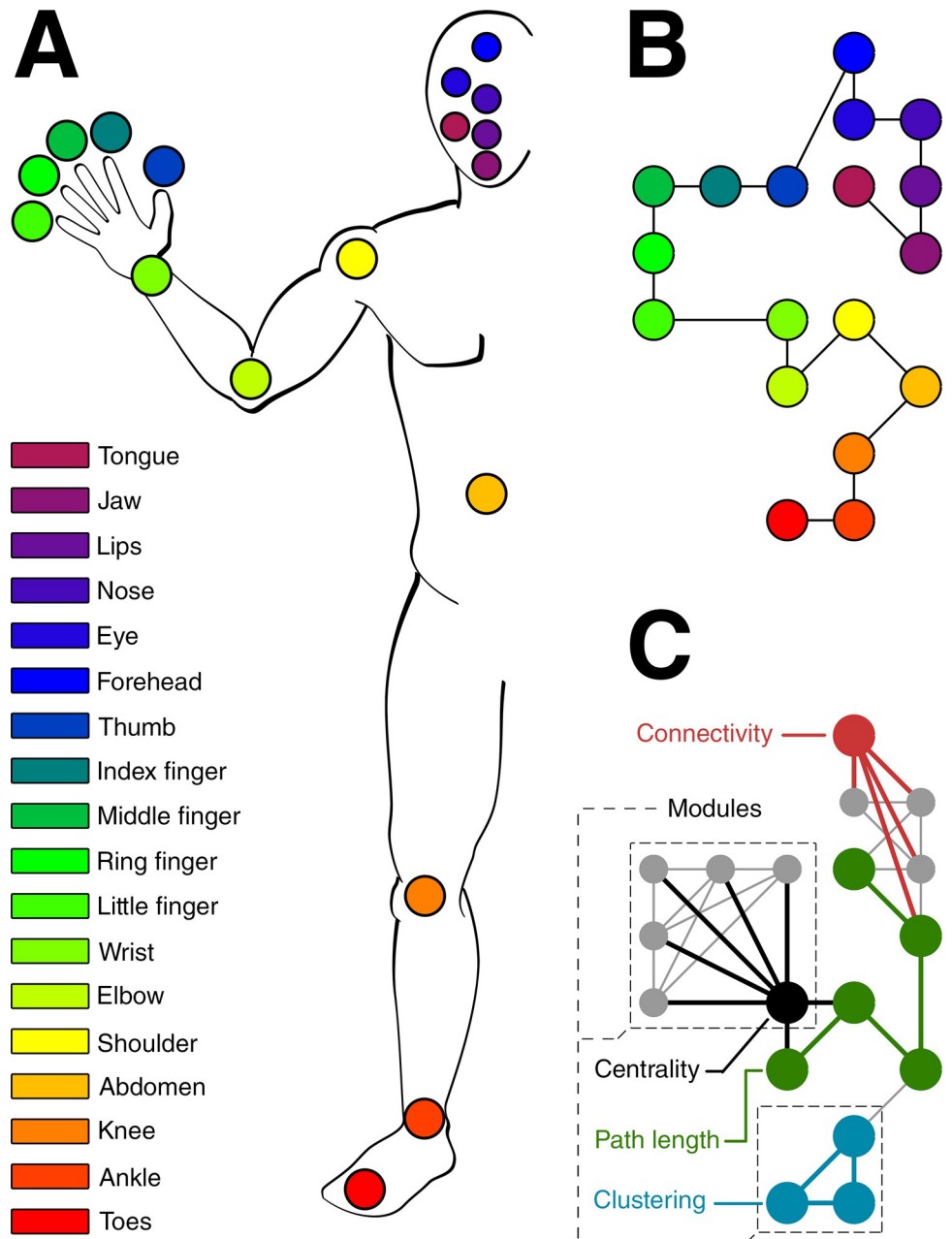

**Fig 1. Body parts and body graphs.** (A) Schematic of the body and the cued body parts (colors) are shown. (B) The layout of the whole body graph is presented with the colored nodes representing the body parts. The position of nodes in the graph is arbitrary and chosen to resemble the physical position of the body parts. The lines denote which body parts are 'connected' on the basis the cortical homunculus ordering of body parts. (C) Schematic of the graph theory metrics: connectivity (red), clustering coefficient (blue), and betweenness centrality (black) that relies on path length (green). Example modules consisting of multiple body part nodes are denoted by the black dashed lines. These graph theory metrics were applied to all body parts in all ROIs. The lines denote existing connections between body part nodes that were determined by correlations of the mean body part pRFs and thresholding. The nodes in the graph have the same order as in (B).

cortical homunculus share a high proximity within response fields. Finally, the graph theory metrics describe the relations of body part representations in different cortical areas from the brain's perspective. The *uniqueness* of body parts is given by the connectivity measure, the

*cliqueness* is given by the clustering coefficient and body part *influence* is given by the betweenness centrality coefficient. The modules reflect which body part response profiles share similar characteristics.

## 2. Results

Eight participants performed an 18-body part motor task, while neuronal population activity was recorded using 7T fMRI. The majority of body part movements entailed a simple flexion-extension movement (Table 1). Motor tasks typically cause more head movement than non-motor task, which potentially affects the quality of the measurements. However, of all movement cues, only the knee movement resulted in significantly larger estimated head motion (Welch $t_{(7)}$ = 3.18, p = 0.015). The mean head displacement of the knee was 2.6mm (S.D. = 1.6mm), while the mean head displacement of other body parts combined was 0.6mm (S.D. = 0.4mm). Therefore, the following non-rigid pRF metrics associated with the knee movement might be affected by relatively increased head motion. Nonetheless, the non-rigid pRF method captures on average 13% of the timeseries variance explained in the cortical areas related to motor control (mean $R^2$ = .13, mean S.D. = .06). Variance explained of the data with the non-rigid pRF model is highest in primary sensorimotor cortex (M1: $R^2$ = .19, S.D. = .07; S1: $R^2$ = .18, S.D. = .09; SMA: $R^2$ = .16, S.D. = .08; PMd: $R^2$ = .16, S.D. = .07; S1: PMv = .12, S.D. = .05; Insula: $R^2$ = .13, S.D. = .05; iPC: $R^2$ = .13, S.D. = .05; sPC: $R^2$ = .14, S.D. = .06. See also S1 Fig for goodness-of-fit statistics).

### 2.1 pRF center

The pRF center reflects the preferred body part for each neuronal population with respect to the 18 body parts that were moved during the fMRI experiment. Representations of all body parts were observed in all included cortical areas (Fig 2). Over 99% (S.D. = .07) of all neuronal populations have a preference for a single body part, and on the basis of the preferred body

**Table 1. The table describes the movements that were made for each body part condition.** Subjects viewed a single forward movement cue and a single backward movement cue per event. Rightmost column indicates in which of the two runs a body part cue was presented.

|     | Body part | Forward movement | Backward movement | fMRI run |
|-----|-----------|------------------|-------------------|----------|
| 1   | Toes | *Flexion* | *Extension* | 2 |
| 2   | Ankle | *Flexion* | *Extension* | 2 |
| 3   | Knee | *Extension* | *Flexion* | 2 |
| 4   | Abdomen | *Muscle contraction/pushing outwards* | *Muscle relaxation* | 1 |
| 5   | Shoulder | *Flexion* | *Extension* | 1 |
| 6   | Elbow | *Flexion* | *Extension* | 1 |
| 7   | Wrist | *Flexion* | *Extension* | 1 |
| 8   | Little finger | *Flexion* | *Extension* | 1 |
| 9   | Ring finger | *Flexion* | *Extension* | 1 |
| 10  | Middle finger | *Flexion* | *Extension* | 1 |
| 11  | Index finger | *Flexion* | *Extension* | 1 |
| 12  | Thumb | *Flexion* | *Extension* | 1 |
| 13  | Forehead | *Muscle contraction/pulling upwards* | *Muscle relaxation* | 2 |
| 14  | Eyelid | *Closing eyelid* | *Opening eyelid* | 2 |
| 15  | Nose | *Flaring nostrils* | *Relaxation nostrils* | 2 |
| 16  | Lips | *Pouting lips* | *Relaxation lips* | 2 |
| 17  | Jaw | *Opening jaw* | *Closing jaw* | 2 |
| 18  | Tongue | *Moving tongue to the right* | *Moving tongue to the center* | 2 |

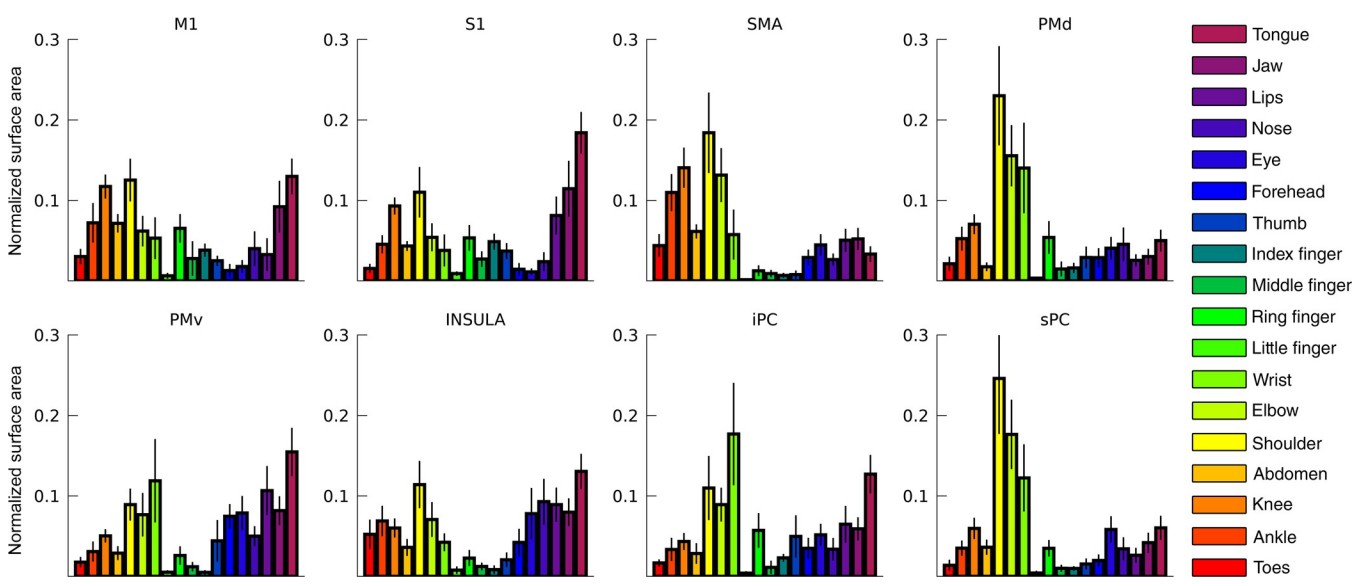

**Fig 2. Surface area.** For each included cortical area, the surface area as estimated by Freesurfer is shown per body part representation. The surface area is normalized to the total surface area of each cortical area. Body part representations are determined on the basis of the pRF center value.

part, somatotopic structures in the left (contralateral) hemisphere can be observed from both a lateral and medial point of view (Figs 3 and S4). Somatotopic structures are most prominent in M1 and S1, reflected by a significant gradual change in preferred body part along the direction of the central sulcus (i.e. pRF center gradients: $t_{(7)} = 20.43$, $p < 0.001$, and $t_{(7)} = 126.77$, $p < 0.001$, for M1 and S1 respectively). Evidence for somatotopic structures is also observed for

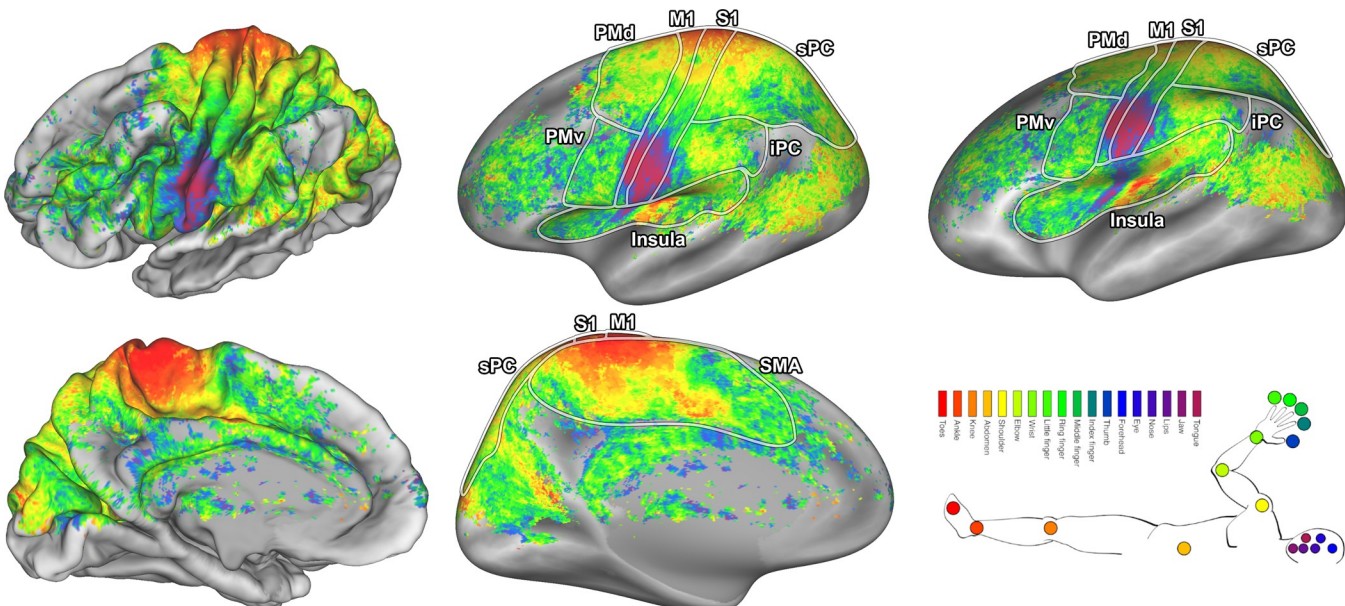

**Fig 3. pRF center maps.** The pRF centers are shown on an average subject pial surface (left) and inflated surface (right) from a lateral point of view (top) and medial point of view (bottom). Colors indicate the body part that was estimated as the pRF center. The ROIs are denoted by the lines drawn on the surfaces: primary motor cortex (M1), primary somatosensory cortex (S1), supplementary motor area (SMA), dorsal premotor cortex (PMd), ventral premotor cortex (PMv), Insula/Sylvian fissure (Insula), inferior parietal cortex (iPC), and superior parietal cortex (sPC).

areas SMA and Insula ($t_{(7)}$ = 4.77, p = 0.002, and $t_{(7)}$ = 8.84, p < 0.001, respectively). Additionally, somatotopic structures are observed in the 4 remaining areas covering premotor and parietal cortex: PMd ($t_{(7)}$ = 7.57, p < 0.001), PMv ($t_{(7)}$ = 4.43, p = 0.003), iPC ($t_{(7)}$ = 3.87, p = 0.006), and sPC ($t_{(7)}$ = 5.56, p < 0.001). For comparison, we have obtained similar pRF center maps using the conventional pRF method (S2 Fig), which correlate significantly with the main pRF center map, derived from the non-rigid pRF model (R = .89, p < 0.001).

## 2.2 pRF size

The pRF size is a single metric that reflects the distribution of body parts within a response field. The unit for pRF size approximates body part density within a response field, which means that a neuronal population with a pRF size of 1 has approximately 1 body part in its response field. The pRF size differs per cortical area ($F_{(7,10)}$ = 23.02, p < 0.001), showing that neuronal populations in M1 and S1 on average have the smallest pRF sizes (Figs 4 and S5). Furthermore, pRF sizes vary depending on the neuronal population's pRF center (Fig 5), showing that a neuronal population's preference for a particular body part affects the population's pRF size ($F_{(17,15)}$ = 28.10, p < 0.001). Neuronal populations that prefer the fingers display relatively large pRF sizes (mean pRF size 5 fingers = 7.69, SD = 1.88), whereas neuronal populations that prefer the knee consistently display smallest pRF sizes (mean pRF size knee = 4.84, SD = 1.46). Without grouping neuronal populations by their preferred body part we observed small-to-large pRF size gradients in SMA ($t_{(7)}$ = 6.69, p = 0.001) and Insula ($t_{(7)}$ = 5.70, p = 0.003, Fig 4), but not in any of the other cortical areas. Finally, the pRF size maps derived from the non-rigid and conventional pRF methods correlate significantly (R = .78, p < 0.001, see S3 Fig), although pRF sizes estimated by the conventional pRF method tend to be larger beyond primary sensorimotor cortices (see S6 Fig). Larger pRF sizes for the conventional pRF method in secondary motor cortices are likely caused by a widening of the Gaussian shape to encompass non-adjacent body parts.

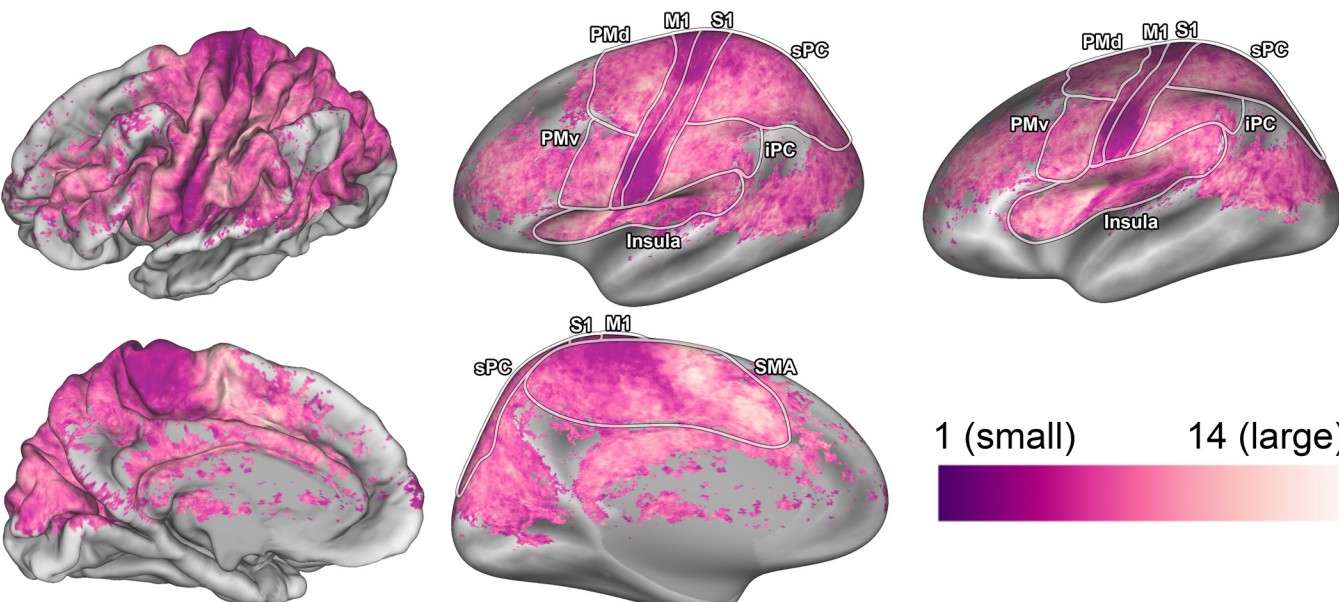

**Fig 4. pRF size maps.** The pRF size is shown on an average subject pial surface (left) and inflated surface (right) from a lateral point of view (top) and medial point of view (bottom). Colors indicate the pRF size. The ROIs are denoted by the lines drawn on the surfaces: primary motor cortex (M1), primary somatosensory cortex (S1), supplementary motor area (SMA), dorsal premotor cortex (PMd), ventral premotor cortex (PMv), Insula/Sylvian fissure (Insula), inferior parietal cortex (iPC), and superior parietal cortex (sPC).

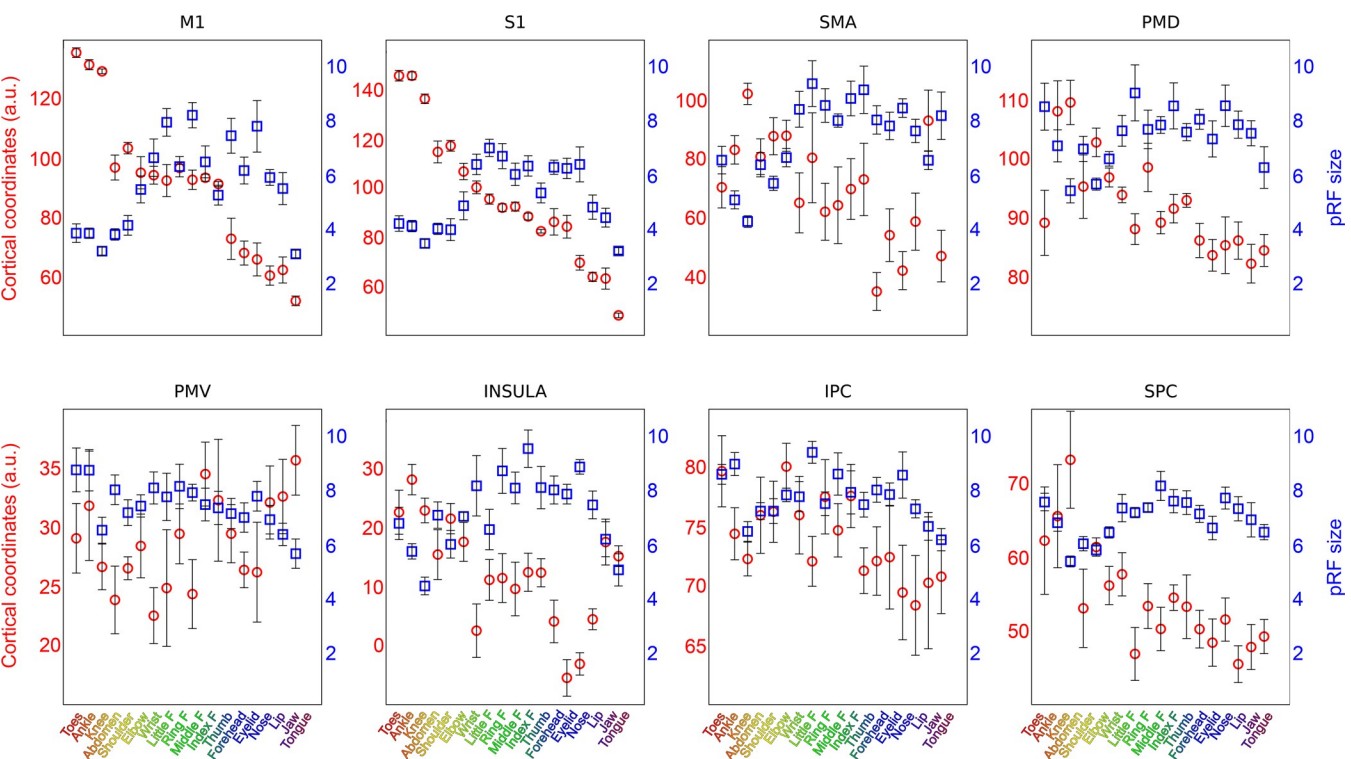

**Fig 5. pRF size and cortical coordinates per pRF center.** For each ROI the mean cortical coordinates (red circles) and mean pRF size (blue squares) are plotted versus the estimated pRF centers (horizontal axis). Both the depicted coordinates and pRF size values were calculated as the mean value across neuronal populations with the same pRF center (horizontal axis). Error bars denote the S.E.M. across subjects.

## 2.3 Response field quantification

The pRF center and size summarize specific aspects of the complete response field. Additionally, we investigate the positioning of all body part representations within response fields. The movement order of body parts was pseudo-randomized to prevent an artificial coupling of body parts on the basis of the experimental design, despite that such artificial coupling may have persisted between separate fMRI runs (Table 1). However, we found no evidence for an artificial coupling of body parts that were cued within a single run across the whole field of view (Welch $t_{(13)}$ = 0.34 p = 0.737, see S1 Text). The full response fields show that for any given body part at the pRF center, if another body part is proximate on the cortical homunculus, it is also proximate to that specific pRF center ($F_{(16,19)}$ = 103.41, p < 0.001). In other words, neuronal populations that have a preference for some body part *P* often contain body parts in their response fields that are adjacent to *P* on the cortical homunculus (Fig 6), thereby revealing a functional adjacency of body part representations in human motor cortices.

The observed functional adjacency of body part representations does not perfectly mirror the cortical homunculus ordering of body parts, especially in cortical areas outside M1 and S1 (Fig 6, e.g. PMv). We use graph theory to investigate the relationships between body part representations. Weighted graphs are constructed for every ROI by correlation of the mean body part pRFs (Fig 6), creating body part nodes and the connections between them. We then extract connectivity, clustering, and betweenness centrality coefficients from the whole-body graphs. We found that connectivity values differed significantly across body parts ($F_{(17,119)}$ = 3.56, p < 0.001) and cortical areas ($F_{(7,49)}$ = 7.85, p < 0.001, Fig 7A). In particular, we found that the knee was less connected within the graphs compared to other body parts ($t_{(119)}$ =

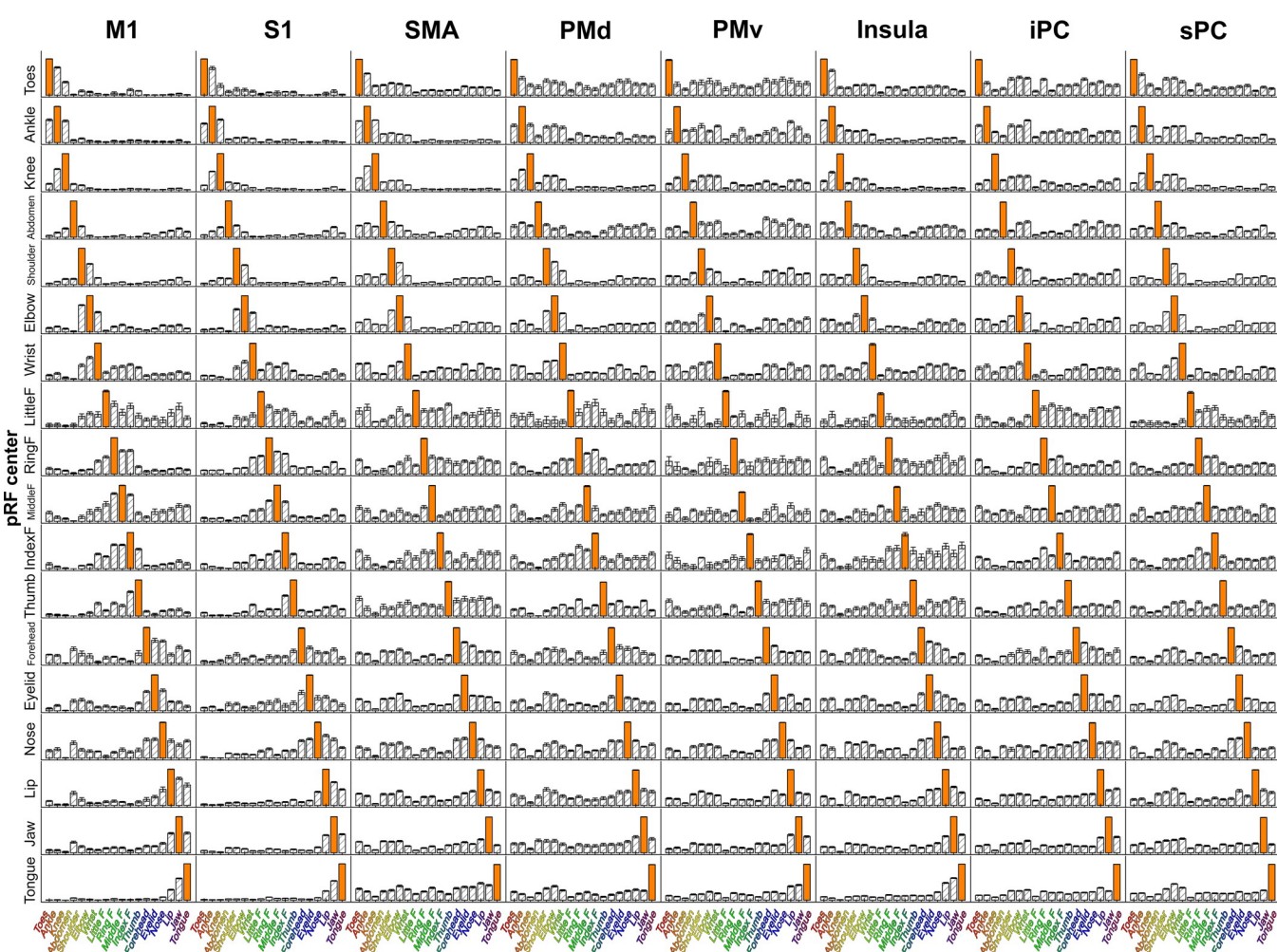

**Fig 6. Mean body part pRF.** For each ROI (columns), the complete response field was normalized and averaged over all vertices sharing the same pRF centers (rows), creating the mean body part pRF (i.e. 18 body part positions per pRF center and ROI). The bars denote the proximity of body parts to the center of the response field. The higher the bar, the closer the corresponding body part is to the response field's center. For each mean body part pRF, the body part equal to the pRF center is depicted by the orange bar, and is by definition closest to the response field center. The Error bars denote the S.E.M. across subjects.

-5.22, p < 0.001). The connectivity averages of cortical areas reveal that body parts in M1 and S1 are less interconnected ($t_{(49)}$ = -4.58, p < 0.001 and $t_{(49)}$ = -4.03, p < 0.001, respectively), while body parts in PMv have above average connectivity values ($t_{(49)}$ = 4.05, p < 0.001). Next, the clustering coefficient differed across body parts ($F_{(17,119)}$ = 2.40, p = 0.003), with the ring finger having significantly larger clustering coefficients ($t_{(119)}$ = 3.04, p = 0.004). No clustering effects were observed across cortical areas ($F_{(7,49)}$ = 1.35, p = 0.249, Fig 7B). Finally, we found that betweenness centrality coefficients differ across body parts and cortical areas ($F_{(17,119)}$ = 2.56, p = 0.002 and $F_{(17,49)}$ = 11.88, p < 0.001, respectively). The shoulder and the wrist exhibited larger betweenness centrality coefficients compared to all other body parts ($t_{(119)}$ = 3.36, p = 0.001 and $t_{(119)}$ = 3.73, p < 0.001, respectively). Additionally, body parts in M1 and S1 contain on average larger centrality coefficients ($t_{(49)}$ = 6.13, p < 0.001 and ($t_{(49)}$ = 5.43, p < 0.001), while average centrality coefficients in PMv and sPC are significantly smaller compared to other areas ($t_{(49)}$ = -3.26, p = 0.002 and $t_{(49)}$ = -2.66, p = 0.010, respectively. Fig 7C).

By viewing body part pRFs as a connected graph we can, additionally, look for modules in the network. Modules are a measure of segregation, but unlike the clustering coefficient, act

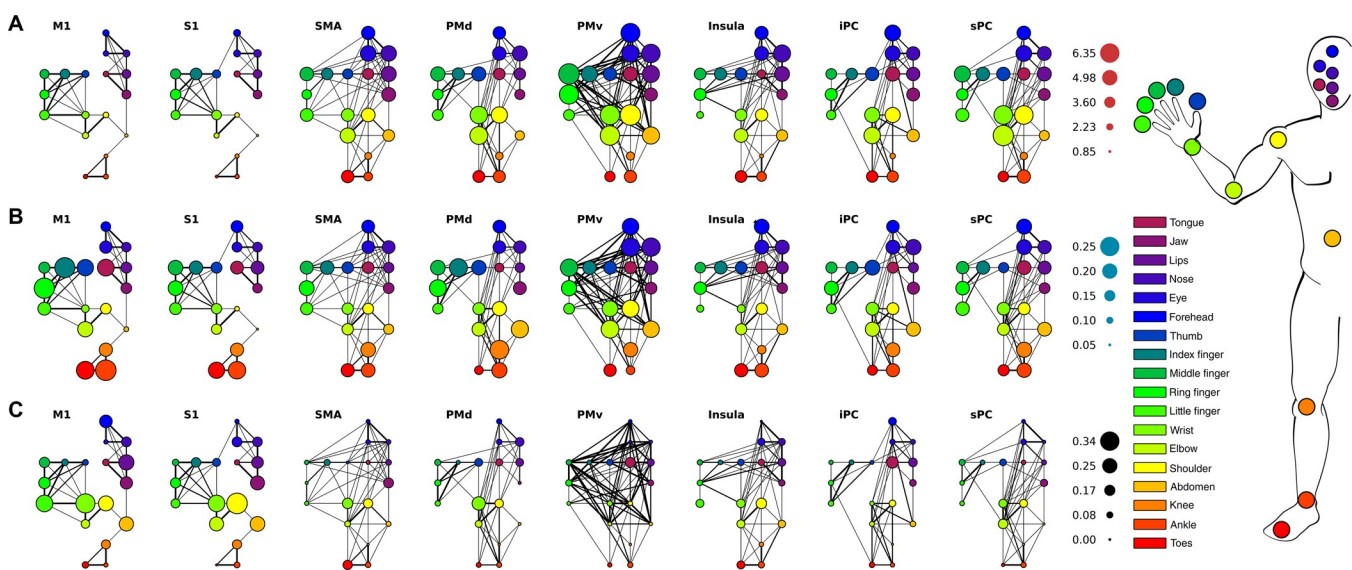

**Fig 7. Graph theory results.** Whole-body-graphs are presented per ROI (from left to right) and for the connectivity, clustering and betweenness centrality coefficients (from top to bottom). The colors of each node in the graphs correspond to a specific body part given by the schematic at the far right. The connections between any 2 body part nodes was calculated per ROI and shown here through the lines connecting the nodes. The thicker the line the stronger the connection between body parts. (A) Connectivity values per body part node and ROI are depicted. The size of the body part nodes presents the size of the connectivity value per node. (B) Clustering coefficients have the same layout and graphs as the connectivity values. Here the size of the body part node reflects the strength of the clustering coefficient. (C) The size of the body part nodes in the ROI graphs reflects the strength of the betweenness centrality coefficient.

on multiple body parts in the network simultaneously. Hence, a qualitative analysis can be performed on the existence of modules consisting of multiple body parts. Body parts within the same module are assigned the same integer value and are indicated by different colors in Fig 8. Using Louvain modularity, we find that particularly in M1 and S1 the cluster assignment of body parts is in agreement with the physical distance of body parts and with co-occurrence of body parts in real-life movements: a toes-ankle-knee cluster; a shoulder-elbow cluster; a cluster of the wrist and the 5 fingers; a forehead-eyelid-nose cluster; and a lip-jaw-tongue cluster (Fig 8). The only difference between M1 and S1 clustering is found at the abdomen, which is clustered together with the bottom half of the face (i.e. lip, jaw, and tongue) in M1, and forms its own cluster in S1. Please note, that the cluster assignments are purely based on the Louvain Modularity method, and their resemblance to somatotopic and physical structuring of body parts emphasizes specific commonalities among response profiles. Clusters derived from the other cortical areas differ each in their own way. Some clusters appeared relatively consistent across the separate brain regions, namely the toes-ankle-cluster cluster (observed in M1, S1 and sPC), the forehead-eyelid-nose cluster (observed in M1, S1, Insula, iPC, and sPC) and the

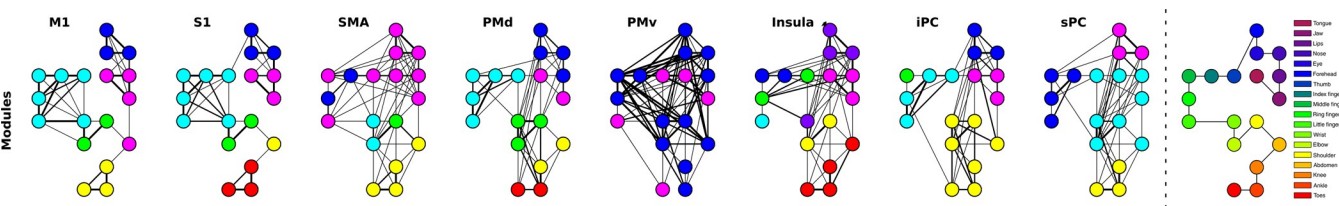

**Fig 8. Body part modules.** For each ROI, different modules are represented by different colors. Note that the colors only define a cluster of nodes within one graph, and any correspondence of colors between graphs is purely accidental. The whole-body graph layout is presented at the outmost right indicating the node-body part relationship.

lip-jaw-tongue cluster (M1, S1, Insula, and iPC). All areas except PMv reveal specific clusters of physically proximate body parts. Area PMv has only two clusters of body parts that do not directly relate to each other from either a physical of functional perspective. For the remaining areas, the combination of non-rigid pRF estimation and graph theory confirms clusters of physically and functionally related body parts and reveal subtle differences in the cortical representation of our body.

## 3. Discussion

### 3.1 General discussion

The aim of the current study was to gain insight into relations between representations of body parts in human sensorimotor cortex. We deployed our novel non-rigid population Response Field (pRF) model to investigate motor cortical activity with fMRI, accompanying an 18-body-part motor task. The pRF centers represent the preferred body part of individual neuronal populations and on the basis of pRF centers we provide strong evidence for previously observed whole-body somatotopic structures in M1, S1, SMA and Insula/Sylvian fissure (Fig 3). In addition, we provide new evidence for the presence of somatotopic structures in other motor-related cortical areas: PMd, PMv, sPC and iPC (Fig 5). In line with expectations, pRF sizes are smaller in primary sensorimotor cortex compared to other frontal and parietal motor areas, indicating that neuronal populations in primary sensorimotor cortex code for fewer body parts than secondary motor cortices (Figs 4 and 5). Non-rigid pRF modeling reveals a high degree of cross-correspondence among body parts. PRF center body parts are frequently neighbored by body parts that are proximate from either a physical or cortical homunculus ordering's perspective (Fig 6). However, the internal structure of response fields is not uniform among body part representations or cortical areas. With the use of graph theory, we find that there are consistent differences in connection strength, clustering and betweenness centrality coefficients among body parts and cortical areas (Fig 7). Furthermore, body part modules can be distinguished on the basis of the mean body part pRFs, revealing that predominantly primary sensorimotor cortex contains modules that are in agreement with the physical proximity of body parts (Fig 8). Thus, the non-rigid pRF model exposes coherent but different functional relations between body part representations across several cortical areas involved in motor functioning.

### 3.2 pRF center and size

The pRF center reflects a neuronal population's preferred body part, which allows for the assessment of somatotopic structures in brain regions. On the basis of previous studies [4,11,12,14,35], we expected somatotopies in M1, S1, SMA, and near the insular cortex. The clearest somatotopies are indeed observed in these areas, although evidence of a somatotopic arrangement is observed for the other included cortical areas as well (i.e. PMd, PMv, iPC and sPC). On average, we find evidence in support for a linear upright somatotopy of head/face representations in M1 and S1, as was originally proposed by Penfield and Boldrey [23]. However, on the basis of visual inspection, the cortical representations of the head and face in M1 and S1 appear to be organized in an approximately polar coordinate system (Fig 3). A similar head/face organization has previously been proposed on the basis of macaque electrophysiology [36] and somatosensory stimulation in humans [37]. One possibility is that the conventional somatotopic head/face organization based on "average cortical coordinates" may be an oversimplification, and that a radial head/face somatotopy might be closer to the truth. Further research is required to evaluate the different somatotopic organizational possibilities.

It needs to be mentioned that, except for M1 and S1, the parcellation of included cortical areas is rather coarse, which likely influences results on somatotopic arrangement. Cortical parcellations that are too large or too small in size risk incorporating multiple or incomplete functionally distinct brain regions, respectively. The current results putatively reveal multiple somatotopic structures in regions SMA and Insula. SMA is known to consist of two separate regions: pre-SMA and SMA proper [38,39], which might each represent the whole body in full. The region denoted insula, which covers the region enclosed by the Sylvian fissure up to the parietal operculum, likely contains the secondary somatosensory cortex (S2), which is believed to contain a whole-body somatotopy as well [40–42]. Further parcellation of these areas might, however, lead to an underrepresentation of certain body parts (see also Fig 2), which could potentially hinder the assessment of somatotopic structures. The selection of regions of interest (ROI) often is an arbitrary and non-trivial process, which could be aided in future studies by the current pRF center results.

The changes in pRF size across the left hemisphere reveal several striking similarities with changes in pRF centers. The outer borders of primary sensorimotor cortex are accompanied by a sudden change in pRF size: primary somatosensory cortex displays relatively small pRF sizes, whereas secondary motor cortices and parietal cortex show substantially larger pRF sizes, indicating that neuronal populations in primary sensorimotor cortex are involved with fewer body parts than other motor related areas. Neuronal populations in M1 have efferent connections to localized motor units and muscle groups [6,15,43–45] and might relate directly to 'muscle fields' demonstrated in animal studies [46,47], whereas S1 activity likely portrays localized proprioceptive feedback information following movements [48,49]. In contrast, pre-motor cortex and SMA are thought to be involved in motor planning and motor sequences, which would yield an integration of multiple body parts that are not necessarily physically connected [2,50–52]. Additionally, small-to-large pRF size gradients are observed in SMA and the insular cortex. The function of these pRF size gradients, and whether they are part of a single functional area or denote the borders between functionally distinct brain regions, remains to be investigated.

With respect to body parts, we found that pRF sizes are largest for the 5 fingers and smallest for the knee. PRF size is a measure of body part representation within neuronal populations and the larger pRF sizes for the fingers indicate that all fingers populate the response fields of all neuronal populations representing the fingers. It is important to state that the estimated pRF sizes are relative to the 18 body parts that were cued for movement. Since more body parts from the upper limb were cued for movement than e.g. the midsection, more upper limb body parts could potentially be included in a response field. Additionally, the finger pRF sizes may be specific to motor-induced activity, and finger pRF sizes solely induced by somatosensory stimulation may be smaller [21,22]. However, others have found no difference in S1 multivariate activity patterns between active motor and passive somatosensory stimulation [53], or increased movement complexity during electrical cortical stimulation of M1 [54]. These previous findings could indicate a difficulty in separating somatosensory and motor activity in vivo. The entanglement of sensorimotor activity is likely also present in our previous finger movement pRF study [18], where we initially showed that neuronal populations coding for fingers frequently incorporate other fingers within their response fields. Thus while fingers are physically small in size, they are often used conjointly and the larger motor-induced finger pRF sizes plausibly reflect their joint (sensorimotor) integration [1,55]. PRF profiles for finger representations might, therefore, by shaped on the basis of their usage [56]. Conversely, the knee is heavily involved in walking and sitting down—actions that are notoriously difficult to test in an MR-scanner—and may actually be integrated with other body parts and movement types that have simply not been tested currently [57]. Moreover, it is probable that neuronal

populations, especially in secondary motor cortices, prefer specific movement types that are not captured by the simple flexion-extension instruction presented here [58–60]. Altering the set of movement types might change the estimated pRF size. Additionally, the knee movement cue led to the largest estimated head movements in the scanner. It is, therefore, possible that increased head motion during scanning might have decreased the estimated pRF sizes.

## 3.3 Body part relationships

Where the pRF center and size present a useful summary of population response fields of individual neuronal populations, the graph theory metrics reveal additional inter-body-part relationships between multiple neuronal populations characterized by connectivity, clustering and betweenness centrality coefficients. The knee exhibited the lowest connectivity values, which means that given the performed movements the knee response profile is relatively unique across all included motor related cortical areas. The low connectivity value for the knee matches the smaller pRF size for neuronal populations, preferring the knee (for the relationship between pRF size and connectivity metrics, see also S7 Fig). The ring finger was found to have a significantly larger clustering coefficient, reflecting the interconnectedness of the ringer finger and the direct neighboring body parts of the ring finger. This interconnectedness only concerns directly connected body parts, while higher-order clustering algorithms (not employed here) might reveal higher-order body part interconnectedness [61]. The large clustering coefficient of the ring finger indicates clique-formation of predominantly the fingers and the upper limb. It is not particularly obvious why the ringer finger is appreciated as the center of clique-formation of the fingers and upper limb, but this might relate to previous findings showing relatively enlarged cortical representations for the ring finger in S1 during somatosensation [62]. One possible explanation might be that the ring finger's constrained freedom of movement [63–65] could result in the ring finger acting as a 'common denominator' for various multi-digit movements, leading to the observed digit interconnectedness surrounding the ring finger [66,67]. On the basis of betweenness centrality coefficients the shoulder and wrist are characterized as central and are, therefore, relatively influential during the movements of other body parts. The involvement of frontal and parietal cortex in upper limb motor control has previously been demonstrated [68–70]. However, current results indicate that shoulder and wrist representations are more influential within the central nervous system, compared to other body parts. Increased levels of centrality for the shoulder and wrist may signify the relative importance of upper limb control within the human motor repertoire from a cortical computation perspective [60,71–73].

The graph theory findings, however, can be influenced by several confounding factors. First, the presentation of body parts was split in two halves (hand/arm/torso and face/leg), which could have caused a coupling of body parts by task design. This could hinder the interpretation of our results. However, we specifically tested for an unintended task design bias, for which we found no evidence when taking into account the whole field of view. Second, the size of the cortical representations of body parts differs in any of the included cortical areas. Therefore, the relationship between body parts is not based on the same number of data points for each body part, and could be skewed towards smaller or larger body part representations. However, when the same analysis is performed while controlling for an equal number of data points per body part, a comparable pattern of body part relationships emerges (see S2 Text).

Averaged over cortical areas, we observe that body parts representations in primary somatosensory cortex (M1/S1) are characterized by relatively unique response profiles and relate fairly directly to the corresponding physical body parts [1,74]. The high correspondence with individual body parts in M1/S1 plausibly relates to the large betweenness centrality

coefficients, signifying that here many body part representations are influential at the scale of individual body part motor control. Additionally, we observed body part modules in M1 and S1 that are in agreement with physical body part adjacency. Since modules are solely based on response profile similarities, they reflect motor planning, motor execution and proprioceptive feedback information [75–77]. The only difference between M1 and S1 is observed for the abdomen, which forms a cluster together with the articulatory body parts lip, jaw and tongue in M1, while forming its own module in S1. Additionally, the wrist is considered part of the fingers in M1 and S1, rather than part of the shoulder and elbow module. The fact that the wrist and shoulder are part of different modules in primary somatosensory cortex combined with their relative increased influence during movements of other body parts (increased betweenness centrality coefficients) could suggest that both body parts act as the leading joint in hand/arm movements relative to their respective module [71,78,79]. Beyond primary sensorimotor cortex, distinctions between body parts are less prominent with PMv showing the least clear body part distinctions. PMv has on average the largest connectivity values and relatively low betweenness centrality coefficients, indicating a lack of body part differentiation. Body part response profiles might be so similar, that it can be disputed if PMv motor calculations involve body part representations at all. Such interpretation is supported by the observed PMv modules, consisting of just two large groups of body parts that do not obviously relate to one another (Fig 8). These findings agree well with the notion that PMv is positioned relatively high up the cortical hierarchy during motor processes, performing abstract rather than body part motivated computations [3,80–82]. Note, however, that several graph theory metrics concerning PMv may be dependent on the included surface area per body part (see S2 Text), which may suggest that area PMv contains a relatively heterogenous collections of neuronal populations. The body part modules that are observed for the remaining cortical areas are a mixture of physically adjacent and distant body part combinations, requiring further research to elaborate on their functions.

### 3.4 Non-rigid pRF model

In sensory cortices, such as visual and auditory cortex, much knowledge has been gained using population Receptive Field modeling and fMRI [19,20,83–86]. However, visual and auditory modalities exhibit a clear continuous relationship between sensation (through the retina and cochlea, respectively) and cortical representation, whereas our body and movements of body parts do not exhibit such an obviously clear connection to cortical representation despite the coarse somatotopic arrangement of corticospinal tracts [87,88]. Population Receptive Field modeling capitalizes on the relation between sensation and cortical representation, but the advantages of population Receptive Field modeling have eluded the sensorimotor system in absence of such relationship. We modified the population Receptive Field model to accommodate the lack of a predefined somatic ordering, while maintaining the ability to extract meaningful features: the non-rigid population Response Field model. Instead of finding the best fit of a Gaussian shaped receptive/response field over a rigid dimension of functional features (e.g. visual field locations, auditory frequencies, body parts), the non-rigid pRF model finds the best fit of functional features within a static Gaussian shaped response field. The rationale for modeling neuronal activity in this manner revolves around the concept of having to—or actually–*not* having to define the reciprocal relation between selected functional features *a priori*. Particularly for cortical motor response fields, we wish not to assume reciprocal relationships between body parts beforehand, since many different body parts can move in conjunction. It needs to be mentioned that we do make the assumption that the shape of the response field is Gaussian. The choice for a Gaussian shape is motivated based on the

successful application of Gaussian response profiles in neuroscientific research [89–91], although other shapes might provide a more accurate display of neuronal functioning. The strength of the non-rigid pRF model lies in its ability to find the correspondence among functional features, quantified as relative distance of functional features from the response field's center or from each other. The positioning of functional features within the response field allows for the assessment of pRF center and size, which correlate significantly with the pRF center and size derived from a conventional pRF model using a rigid cortical homunculus-like ordering of body parts. There are, however, several noteworthy differences between the non-rigid and conventional pRF models: the conventional pRF model returns a center value on a continuous, rather than discrete, feature dimension. It can, therefore, return a fractioned value for the pRF center, allowing the center to be positioned between two body parts. Furthermore, the pRF sizes were on average larger for the conventional compared to the non-rigid pRF model (except for M1 and S1). This finding may reflect that when response fields become larger and/or more complex (i.e. beyond primary sensorimotor cortices), the conventional pRF model is forced to widen its Gaussian shape to encompass body parts that are not directly adjacent to each other with respect to the cortical homunculus ordering of body parts (e.g. hand-mouth response fields in parietal cortex [92]).

We have not compared our non-rigid pRF model with the commonly used general linear model (GLM) approach. However, from a theoretical stance we argue that it does not differ with respect to estimated amplitude per functional feature (i.e. per condition), or the explained variance by either model. The current study suffered from a relative low variance explained mainly caused by a poor fit between the canonical HRF and observed BOLD response. However, the GLM approach does differ with respect to the correspondence between features. A standard GLM returns regression coefficients per feature and binary statistical tests can be performed to identify significantly deviating signals [93–95]. However, a GLM does not inform on the correspondence or clustering of features within a neuronal population. One thing the non-rigid pRF method does have in common with a standard GLM-analysis is that it can be readily applied to any set of features. Within-feature correspondence can be assessed with "Representational Similarity Analysis (RSA), which bares several similarities to our non-rigid pRF method [96]. Both methods are well-suited to investigate the relation within sets of features, although the non-rigid pRF method primarily returns response field properties (i.e. pRF center and size), plausibly reflecting afferent connections. Thus, when the intrinsic structure of a response field with respect to a set of features is unknown or subject to investigation, the non-rigid pRF method provides information on cross-correspondence among features within neuronal populations, through which metrics such as pRF center and size and even complex feature networks can be derived.

## 3.5 Conclusions

Accompanying an 18-body-part motor task, we present evidence for somatotopic organizations in cortical areas M1, S1, SMA, PMd, PMv, Insula, iPC and sPC on the basis of non-rigid pRF centers. Additionally, non-rigid pRF sizes vary across the contralateral hemisphere, showing that neuronal populations in M1 and S1 are involved with fewer body parts compared to the other cortical areas. Furthermore, our novel non-rigid pRF method reveals exactly how all 18 body parts are represented in each neuronal population's response field. Using graph theory we were able to define the relationship between body parts in human motor cortex, revealing that the knee is represented by a relatively unique response field, digits of the hand cluster around the ring finger, and that the shoulder and elbow occupy a relatively influential role in motor cortex. The novel non-rigid pRF model together with graph theory network

quantification provides a powerful tool for investigation of neuronal population response fields, when the relationship among selected functions is not known beforehand.

## 4 Material and methods

### 4.1 Ethics statement

Eight healthy volunteers (mean age = 24 years, s.d. = 2.2 years, female = 3, right-handed = 8) participated in this study. All participants gave written informed consent before entering the study. The protocol was approved by the local ethics committee of the University Medical Center Utrecht, in accordance with the Declaration of Helsinki (2013).

### 4.2 Motor task

The participants carried out a movement task in a Philips 7 Tesla MRI scanner that required the separate movement of 18 different body parts. Instructions were projected on a screen in the scanner bore, which were viewed through prism glasses. The movement cues were presented using two different screen images: one image showing the torso, arm and hand, and another image showing a face on the left and a foot/leg on the right. Each of these images was used to present 9 different motion cues (Fig 9). When the movement of any body part was cued, a green circle was presented for 1 second over the cued body part, at which point the participant was to move the cued body part to an instructed position. One second later a red circle was shown for 1 second, during which the participant moved the cued body part back to the starting position. For most body parts the movement procedure meant a single flexion movement, followed by an extension movement. However, there were exceptions such as the abdomen, forehead and eyelid movement instructions (see Table 1). Furthermore, since participants were supine and the knee was supported with a cushion, the knee was first extended and then flexed. Each motion cue was repeated 9 times and the movement order was pseudo-randomized, to prevent systematic sequences. The inter cue interval was 10 seconds, except for 1 randomly chosen repetition per condition, when the interval was lengthened to 14.7 seconds. The participants practiced the task several times outside the scanner until they felt comfortable with the task.

### 4.3 Image acquisition

Scanning was performed on a 7 Tesla Philips Achieva scanner (Philips Healthcare, Best, Netherlands) with a 2-channel volume transmit coil and a 32-channel receive headcoil (Nova Medical, MA, USA). Functional MRI (fMRI) measurements were obtained using a whole-brain echo-planar imaging (EPI) sequence with the following parameters: SENSE factor = 3.5, TR = 2100 ms, TE = 27 ms, flip angle = 70˚, axial orientation, interleaved slice acquisition, FOV (AP, FH, LR) = 208.8 x 41.6 x 208.8 mm3. The acquired matrix had the following dimensions: 132 x 26 x 132, voxel size: 1.75 x 1.75 x 1.75 mm3. The functional session was split in 2 parts (torso/arm/hand and head/leg, Fig 9). These 2 parts were recorded during 2 separate runs each (run I: 237 functional scans, run II: 191 functional scans), resulting in 428 functional scans for each part (i.e. 856 functional scans for both parts) per participant. For participant 1, the reconstruction of run II failed at the scanner. Therefore, only run I was analyzed for this participant. Following the functional sessions, a T1-weighted volume of the whole brain (0.8 x 0.8 x 0.8 mm3, FOV = 238 x 238 x 238) and a whole-brain proton density volume (0.98 x 0.98 x 1.0 mm3, FOV = 256 x 256 x 190) were acquired.

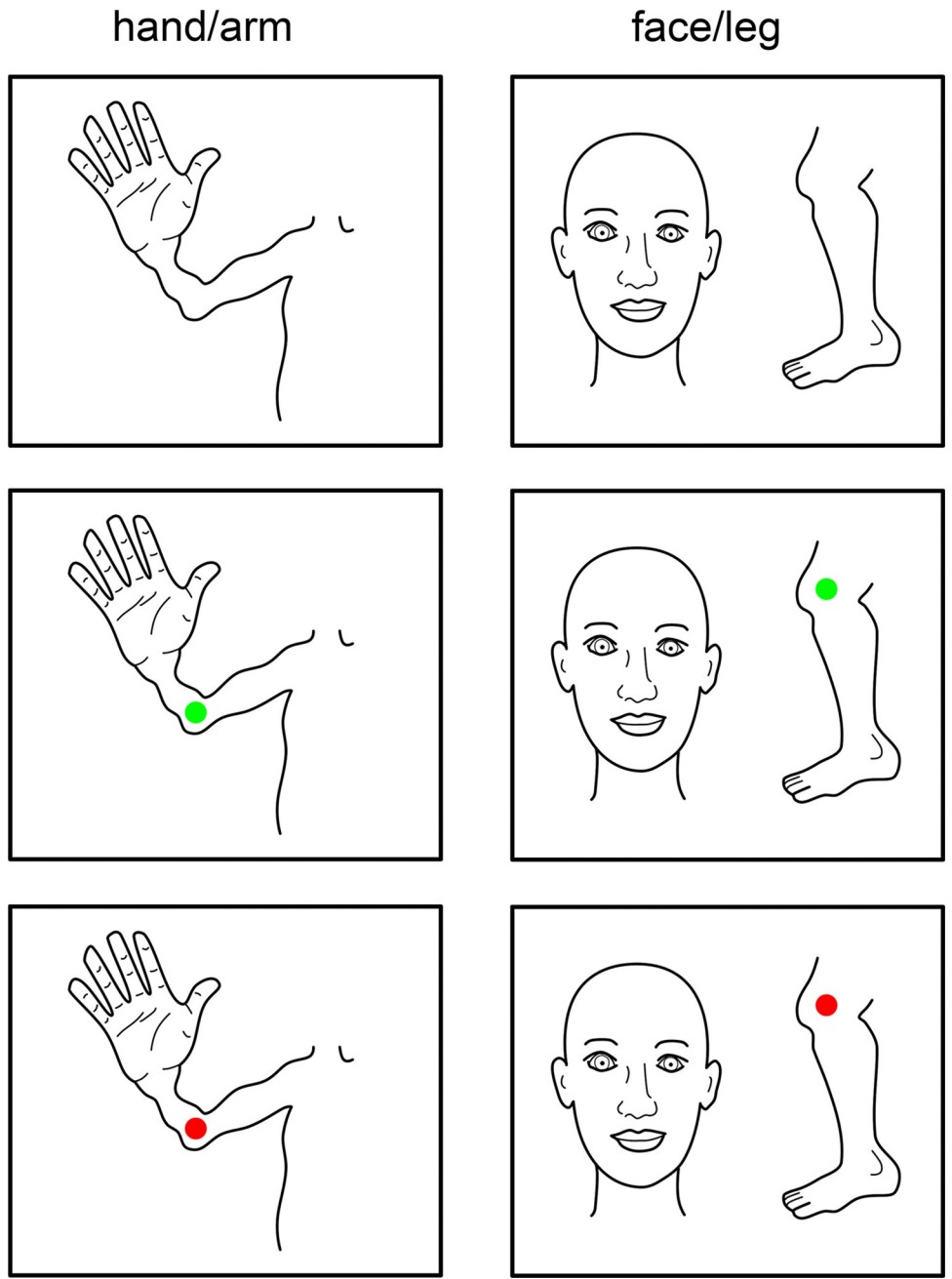

**Fig 9. Scanner instructions.** Top row shows the images presented throughout the task for the hand/arm/torso run (left) and face/leg-run (right). Middle and bottom rows show examples of the flexion-extension cues for the elbow (left) and knee (right).

### 4.4 Image processing

The T1-weighted volume was divided by the proton density volume in order to correct for macroscopic field inhomogeneities present in the T1-weighted volume [97]. The corrected T1-weighted volume was used to construct 3D surface meshes of the white matter and pial surfaces using Freesurfer (http://surfer.nmr.mgh.harvard.edu/, [98]). The functional volumes were corrected for slice time acquisition, head movements, and geometric distortions using

the FSL functions Slicetimer, Mcflirt, and Top-up, respectively [99]. Afterwards, the preprocessed functional volumes were projected onto the Freesurfer surface meshes, where only those voxels were selected that overlapped with estimated cortical grey matter. This procedure resulted in a timeseries per surface vertex (sometimes also referred to as 'surface nodes'). The timeseries from both runs (i.e. torso/arm/hand and head/leg) were concatenated, high-pass filtered with a cut off at 0.01 Hz and rescaled to percent BOLD signal change.

The non-rigid and conventional pRF analyses (see below) were performed in subject space (i.e. on the surface mesh generated per subject). However, the pRF analyses' output was projected on an average subject surface mesh generated with Freesurfer. The average subject surface mesh was also used to draw regions of interest (ROI). We used the Brodmann area atlas supplied by Freesurfer [100] to draw the borders of M1 (BA4a, BA4p) and S1 (BA3a, BA3b, BA1). The borders of other cortical areas were less strictly defined, although PMd, PMv, iPC and sPC were primarily based on the Destrieux atlas [101], while areas SMA simply covered the medial side of the left hemisphere and area insula covered the cortical region enclosed by the lateral sulcus ranging from frontal to parietal operculum. These regions were selected specifically for their relation to motor control.

## 4.5 Non-rigid pRF analysis

For the main analysis, we developed a novel population Response Field (pRF) model that does not assume reciprocal relations between body parts, unlike more conventional pRF methods. The non-rigid pRF model does not try to fit a Gaussian function over a rigid functional dimension of e.g. body parts, rather it keeps the Gaussian function constant and finds the best fit of body parts within:

$$g(x_i) = exp\left(-\frac{(x_0 - dx_i)^2}{2 \cdot \sigma^2}\right), x_i \in N, x_0 = 0, \sigma = 1, dx_i \in \{\mathbb{R}_{\geq 0} | \mathbb{R}_{\leq 10}\} \tag{1}$$

Where $N$ is the list of body parts indexed from 1 to 18. Parameters "$x_0 = 0$" and "$\sigma = 1$" are the center and size of the Gaussian response field respectively and are, thus, held constant. The placing of the body parts within this Gaussian shape is controlled through $dx_i$, which denotes the distance of each body part $x_i$ to the center of its response field ($x_0 = 0$). This means that a body part with a distance of $dx_i = 0$ is at the center of a neuronal population's response field. The larger $dx_i$ becomes, the further away it is from the response field's center. Please note, that we only fit body parts in 1 side of the Gaussian function (i.e. positive values only). We could have allowed $dx_i$ to be negative, but with a symmetrical Gaussian it would have had no effect on the estimated fit, but it does imply a left/right-hand side relationship that we cannot verify. Hence, values of $dx_i \geq 0$ are accepted during fitting. Furthermore, there was a limit applied to the maximum value of $dx_i = 10$ at which point the static Gaussian function with a standard deviation of 1 has a value of near zero. The limit prevented conditions from reaching unnecessarily large $dx_i$ values. Afterwards, the Gaussian function $g(x_i)$ was multiplied by the 2-dimensional movement task design matrix (body parts * time) and summed over the body parts (2):

$$r(t) = \sum_{i \in N} s(x_i, t) \cdot g(x_i) \tag{2}$$

Where $r(t)$ is the effective timeseries, $s(x_i, t)$ is the movement task design matrix and $g(x_i)$ is the non-rigid Gaussian model. The effective timeseries r(t) is then convolved with a canonical hemodynamic response function (HRF) (3):

$$p(t) = r(t) * h(t) \tag{3}$$

Where $p(t)$ is the predicted timeseries, $r(t)$ the effective timeseries and $h(t)$ is the canonical HRF. Finally, the predicted timeseries $p(t)$ is compared with the observed fMRI timeseries $y(t)$:

$$y(t) = \beta \cdot p(t) + \epsilon \tag{4}$$

Where $y(t)$ is the observed fMRI timeseries of a given vertex, $p(t)$ is the predicted non-rigid pRF timeseries, $\beta$ is a scalar, and $\varepsilon$ is measurement noise. We used the Levenberg-Marquardt algorithm (LMA), which is the least-square minimization algorithm [102] used to find the best parameter fits (Fig 10).

Each vertex '$v$' was assigned a pRF center, which was the index of the body part (ranging from 1 to 18) with the lowest distance to the center $x_0$. In case of multiple body parts with the lowest distance to $x_0$ (which occurred in less than 1% of all vertices), the pRF center was calculated as the mean index of the multiple body parts and rounded to the nearest whole integer:

$$pRFC(v) = D(dx_v) \tag{5}$$

$$D(dx_v) = \begin{cases} \min(dx_v), |\min(dx_v)| = 1 \\ \lfloor \dfrac{\sum_{i \in \min(dx_v)} i}{|\min(dx_v)|} \rfloor, |\min(dx_v)| > 1 \end{cases} \tag{6}$$

Where $pRFC(v)$ is the pRF center for vertex $v$, $dx_v$ is the 18-element array of body part distances to the Gaussian response field center $x_0$ estimated for vertex $v$. The function '$min()$' returns the minimum value of an array and '$|min()|$' returns the cardinality of elements with the lowest value. The pRF size was estimated as the sum of normalized distances ($P(dx_v)$) of body parts that were in range of the full-width-at-half-maximum (FWHM) of the response field:

$$pRFS(v) = \sum P(dx_v) \cdot FWHM/2 \tag{7}$$

$$P(dx_v) = \begin{cases} \dfrac{(-dx_v + dx_{max})}{dx_{max}}, dx_v \leq FWHM/2 \\ 0, dx_v > FWHM/2 \end{cases} \tag{8}$$

Where $dx_v$ is the 18-element array of relative distances to the response field center $x_0$ estimated at vertex $v$. $dx_{max}$ is the maximum value that $dx_i$ could attain (i.e. $dx_{max} = 10$), and with a static Gaussian standard deviation of $\sigma = 1$ the FWHM$\approx$2.355. Since body parts were only fitted in one side of the Gaussian shaped response field, the pRF size is calculated as the spread of body parts within the half width at half maximum (i.e. $dx_v \leq FWHM/2$, see also Fig 11).

We, additionally, performed the conventional pRF analysis for comparison purposes. The only difference with the non-rigid pRF model is the Gaussian model function and the parameters that are fitted with the LMA. Instead of function (1), function (9) is inserted in the pipeline:

$$g(x_i) = \exp\left(-\frac{(x_0 - x_i)^2}{2 \cdot \sigma^2}\right), x_{i \in} N, x_0 \in \{\mathbb{R}_{\geq 1} \mathbb{R}_{\leq 18}\}, \sigma \in \{\mathbb{R}_{>0}\} \tag{9}$$

Here $x_i$ is the rigid indexation of a body part (1 = toes, 18 = tongue) and is not updated during the fitting procedure. Parameter $x_0$ is the pRF center, and $\sigma$ denotes the pRF size of the neuronal population. The best model fit for the conventional pRF model was obtained by continuously updating parameters $x_0$ and $\sigma$ from Eq (9).

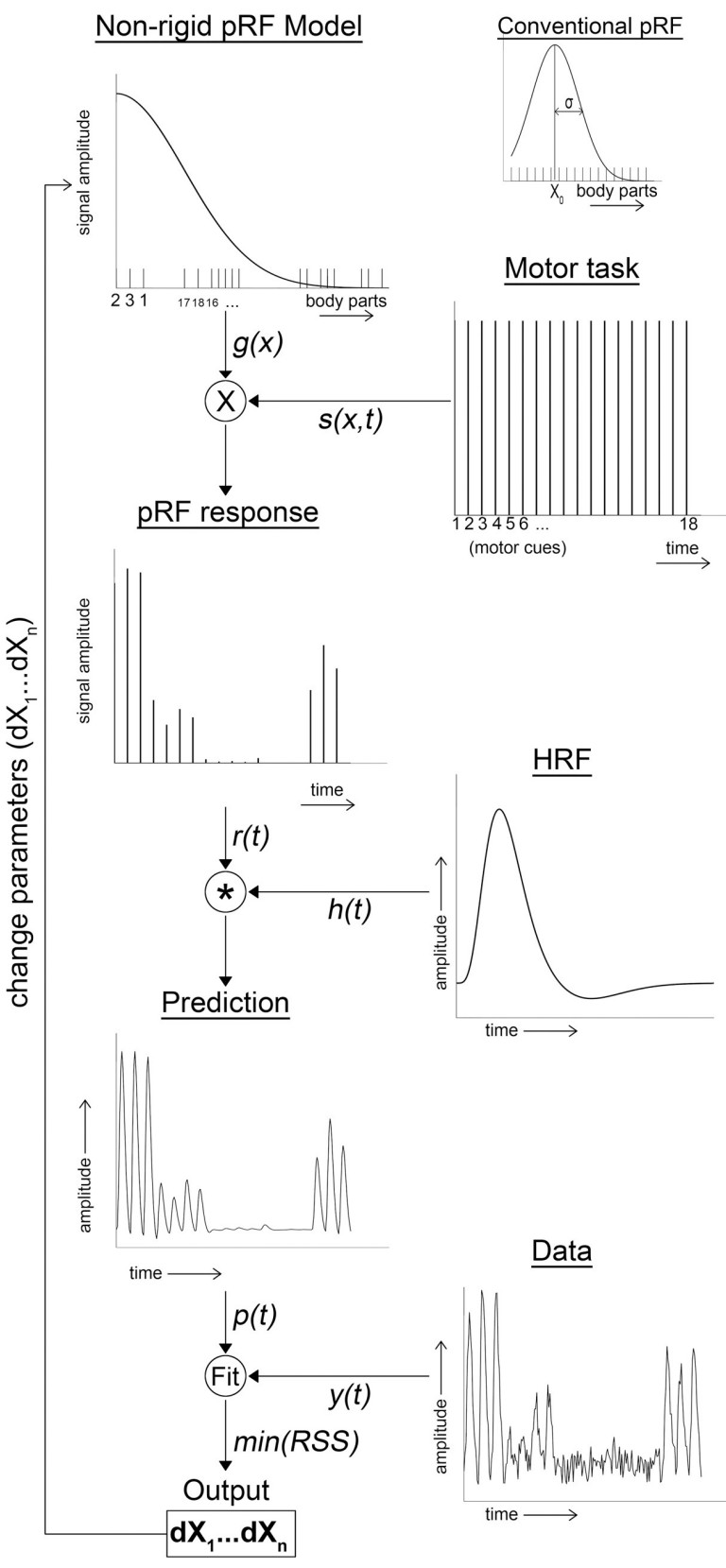

**Fig 10. pRF fitting procedure.** The schematic shows the different steps in the fitting procedure. The different body parts are fitted in one half of a static Gaussian response field model (non-rigid pRF model), which is multiplied by the motor task onset design matrix (Motor task). The multiplication generates the estimated pRF response amplitude for each condition in time (pRF response), which is convolved with a canonical hemodynamic response function (HRF). This results in predicted timeseries (Prediction), which is contrasted with the observed fMRI timeseries (Data). Using the LMA, the position of the body parts in the non-rigid pRF model is updated in order to obtain the best fit.

## 4.6 Graph theory

The non-rigid pRF analysis returns the following parameters for each surface mesh vertex $v$: pRF center, pRF size, and the distance of all 18 body parts to the response field center ($dx_v$). For each vertex a goodness-of-fit F-statistic was calculated for the obtained pRF fit with respect to the measured vertex' fMRI timeseries. Only vertices showing a significant goodness-of-fit F-statistic, false discovery rate (FDR) corrected, were selected for further analyses and were mapped to the average subject surface mesh. Per cortical area, the mean response field of the 18 body parts was calculated as follows:

$$\mu dx_i = \sum_{v \in V_i} \frac{P(dx_v)}{|V_i|}, \{v \in V_i | pRFC(v) = i\} \tag{10}$$

Where $\mu dx_i$ is the averaged 18-element array of normalized distances ($P(dx_v)$) of a set of vertices ($V_i$) where the pRF center is equal to body part $x_i$ ($pRFC(v) = i$). Thus, the averaged distances array $\mu dx_i$ is calculated for each cortical area and each body part being at the center. This results in 8x18 (ROIs x body parts) average response fields, each containing the relative distance of the 18 body parts (see also Fig 6).

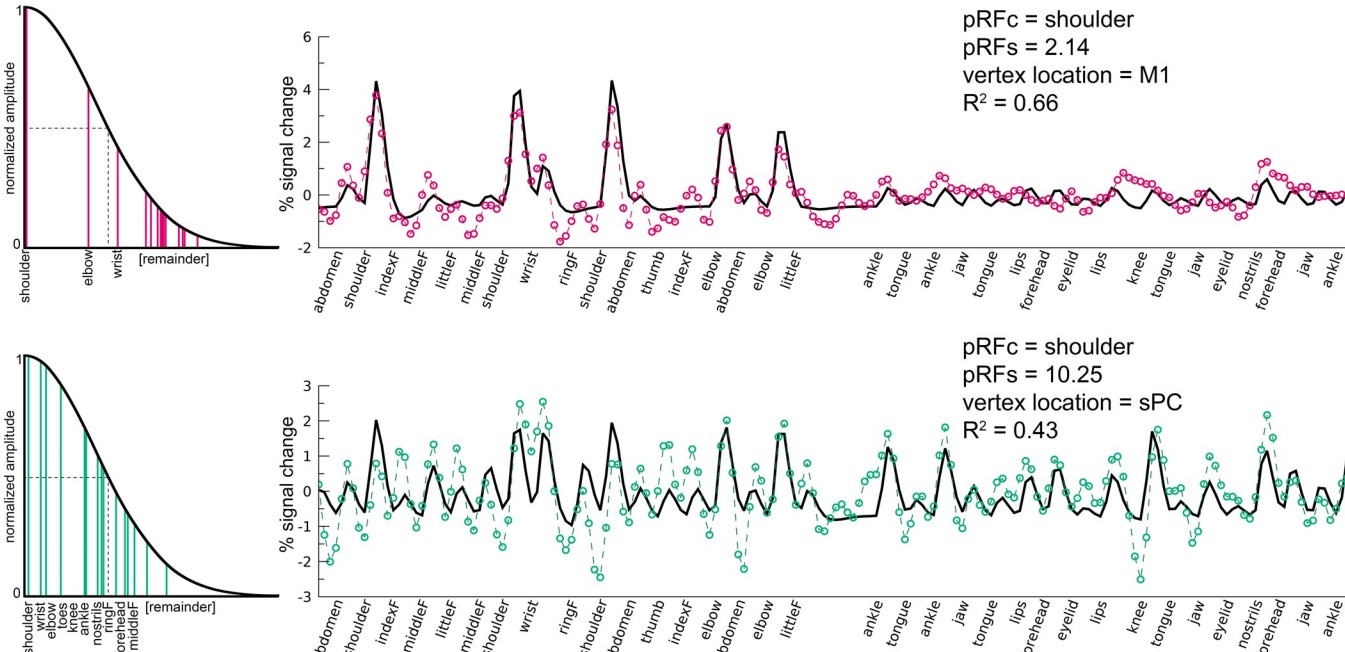

**Fig 11. Non-rigid pRF timeseries.** The non-rigid pRF fit (black line) with the timeseries of 2 surface vertices (colored dashed lines) are shown for 1 participant (#6) in the upper and lower panels. Movement cues over time are presented at the horizontal axis. On the right, the fit of body parts within the static Gaussian response field is shown, used to construct the timeseries fit. The FWHM/2 within the static Gaussian response field is depicted by the black dashed line. The pRF size is determined on the number and spread of fitted body parts (mentioned on the horizontal axis) within the FWHM/2.

To construct weighted graphs of body parts, we calculated the 18x18 correlation matrix of average response fields of each body part for each ROI separately. The Pearson correlation coefficient served as the connection strength, or weights, between body parts. As a final step, the correlation matrix needed to be thresholded to remove low and negative correlation coefficients. This is an arbitrary process, and we chose to include only positive values with a cut-off at the lowest 5% of correlation coefficients per cortical area. Disregarding negative and the lowest 5% of positive correlation coefficients resulted in connected graphs without islands, while removing spurious connections [24,103]. At the end of this procedure, we have 8 (for each ROI) weighted graphs with 18 nodes each. The 18 nodes represent the body parts. Whether or not node $i$ was connected to node $j$, thus, depended on the correlation of average response fields $\mu dx_i$ with $\mu dx_j$.

Graph theory offers many metrics, of which some represent similar ideas. We chose the commonly used metrics Connectivity, Clustering and Betweenness Centrality as measures of body part information distribution. All graph metrics were calculated per subject, per graph (i.e. ROI), per node (i.e. body part representation). The connectivity of node $i$ in a weighted graph is equal to its weighted degree ($k$):

$$k_i^w = \sum_{j \epsilon N} w_{ij} \tag{11}$$

Where $N$ is the set of nodes in the graph (i.e. 18 body parts), and $w_{ij}$ is the weight between node $i$ and node $j$. The weighted clustering coefficient measures segregation of nodes from the network. First, the weighted geometric triangles ($t$) around node $i$ are calculated:

$$t_i^w = \frac{1}{2} \sum_{j,h \in N} \left( W_{ij} W_{ih} W_{jh} \right)^{\frac{1}{3}} \tag{12}$$

The weighted clustering coefficient ($C$), then is calculated by normalization of the weighted geometric triangles around node $i$ using the weighted degree ($k$).

$$C_i^w = \frac{1}{n} \sum_{i \in N} \frac{2 t_i^w}{k_i^w (k_i^w - 1)} \tag{13}$$

Finally, the betweenness centrality is assessed. The betweenness centrality coefficient reflects the centrality of the position of nodes in the network on the basis of path length and the fraction of shortest paths passing through a node. First, the shortest weighted path length between any two nodes is assessed:

$$d_{ij}^w \sum_{a_{uv} \epsilon g^w{}_{i \leftrightarrow j}} f\left( W_{uv} \right) \tag{14}$$

Where $f(W_{uv})$ is a mapping function from weight to length. In our study the inverse of the weight was used. Then, $g^w{}_{i \leftrightarrow j}$ is the shortest weighted path between nodes $i$ and $j$. The shortest weighted path was found through an extensive search of each graph. Now with the shortest weighted path length, the weighted characteristic path length ($L$) is assessed for each node:

$$L_i^w = \frac{1}{n} \sum_{i \epsilon N} \frac{\sum_{j \epsilon N, j \neq i} d_{ij}^w}{n - 1} \tag{15}$$

The betweenness centrality coefficient ($b$) for node $i$ is then calculated as follows:

$$b_i = \frac{1}{(n-1)(n-2)} \sum_{\substack{h,j \in N \\ h \neq j, h \neq i, j \neq i}} \frac{\rho_{hj}(i)}{\rho_{hj}} \qquad (16)$$

Where $\rho_{hj}$ is the number of shortest paths through nodes $h$ and $j$, and $\rho_{hj}(i)$ is the number of shortest paths through nodes $h$ and $j$ that also pass through node $i$. Last, Louvain modularity was calculated as follows:

$$Q^w = \frac{1}{l^w} \sum_{i,j \in N} \left[ w_{ij} - \frac{k_i^w k_j^w}{l^w} \right] \delta_{m_i, m_j} \qquad (17)$$

Where the network is fully subdivided in $m$ modules, $m_i$ is the network containing node $i$, and $\delta m_i, m_j = 1$ if $m_i = m_j$ and $\delta m_i, m_j = 0$ if $m_i \neq m_j$. $w_{ij}$ is the number of edges between nodes $i$ and $j$ and $l_w$ is the total number of edges in the graph. Modular structures are found by iteratively optimizing $Q^w$.

## 4.7 Statistical analysis

Head motion parameters were estimated during the volume realignment with FSL Mcflirt. These motion parameters were used to assess head motion following each of the 18 motor cues. The first derivative of the 6 rigid body motion parameters (3 x translation and 3 x rotation) were calculated and summed over the first 3 seconds following a motor cue. Using a Welch t-test, we tested if any of the movement cues caused a significant increase in estimated head motion relative to other movement cues. No data was removed on the basis of estimated head motion.

The first test of fMRI data verified the presence of somatotopic structures. A somatotopic structure is defined as a series of cortical body part representations that show a gradual shift in cortical location. Only surface vertices with a significant goodness-of-fit F-statistic (FDR-corrected) were selected for the statistical analyses (see also S3 Fig). The pRF center value was used for this together with the x/y-coordinates of the flattened average subject cortical surface mesh. For the majority of ROIs, it was not *a priori* known in which cortical direction, if any, a somatotopy could be observed. To try to account for that, we automatically rotated each ROI, so that the mean coordinate of vertices having the toes, ankle, or knee as pRF center pointed south (i.e. low vertical coordinates) and the mean coordinate of vertices with the lip, jaw, or tongue as pRF center pointed north (i.e. high vertical coordinates). Using a linear regression per subject and ROI on the pRF center versus rotated vertical coordinates, we assess the existence of somatotopic structures: regression coefficients significantly larger than zero indicate a gradual increase of pRF center with rotated vertical coordinates, which was tested with a student's t-test across subjects per ROI. We carried out the same linear regression analysis with pRF size over rotated vertical coordinates to test for gradual changes in pRF size per cortical location (i.e. pRF size gradients). Additionally, pRF size was also analyzed using a repeated measures analysis of variance (ANOVA). The average pRF size per pRF center was calculated first: i.e. an average of pRF size values across voxels with the same pRF center per ROI per subject. The pRF center and ROIs were added as factors in the repeated measures ANOVA, allowing us to test for the effects of pRF center and ROI on pRF sizes across subjects.

The last analyses performed on the common pRF metrics center and size were a series of correlation analyses. In the first analysis, the non-rigid pRF centers were correlated with the non-rigid pRF sizes. The pRF center and size maps were averaged over the number of subjects, creating 2 single maps with a value per surface vertex. These maps were correlated with each

other using Pearson correlation, resulting in a single correlation value of which the statistical significance was calculated with the number of non-zero surface vertices minus 2 as the degrees of freedom (significance threshold was Bonferroni corrected). Similarly, correlation values were calculated between the averaged non-rigid pRF center map and the averaged conventional pRF center map, and the averaged non-rigid pRF size map with the averaged conventional pRF size map.

For the average distance of body parts to the center of the response field ($dx_v$), we first tested if the estimated distances were affected by task design. For each vertex, we divided the body part distances into two groups: either presented in the same run as the pRF center of the vertex, or presented in the other run. Using a Welch t-test, we tested if body part distances differed depending on the experimental design. Next, we tested whether there was an effect of body part (i.e. the cortical homunculus ordering) on distance to the center of the response field. For each body part being at the response field center, we averaged the distance of body parts that were one step away on the homunculus ordering. Then, the distance of body parts that were two steps away, and so until the maximum ordering distance of 17 body parts was reached. Using a repeated measures ANOVA with *a priori* specified linear contrast we tested if the distance from the response field center would increase (linearly) with increasing distance regarding the cortical homunculus.

Finally, we tested for significance of all graph theoretical metrics (connectivity, clustering and betweenness centrality coefficients) separately, using a 2-way repeated measures ANOVA per metric with nodes (body parts) and ROI as factors. Additionally, deviation contrasts were defined beforehand, testing for significant differences of any node's or ROI's metric compared to the averaged corresponding metric of all other nodes or ROIs. All statistical tests were performed using *JASP* (https://jasp-stats.org).

## Supporting information

**S1 Fig. Non-rigid pRF goodness-of-fit F-statistic.** Goodness-of-fit F-statistic for the non-rigid pRF method is displayed on an average subject pial surface (left) and inflated surface (right) from a lateral point of view (top) and medial point of view (bottom). Hypothesis and error degrees of freedom for the calculation of the F-statistics were 19 and 837, respectively. The ROIs are denoted by the lines drawn on the surfaces: primary motor cortex (M1), primary somatosensory cortex (S1), supplementary motor area (SMA), dorsal premotor cortex (PMd), ventral premotor cortex (PMv), Insula/Sylvian fissure (Insula), inferior parital cortex (iPC), and superior parietal cortex (sPC).
(TIF)

**S2 Fig. Conventional pRF center maps.** The conventional pRF centers are shown on the average subject pial surface (left) and inflated surface (right) from a lateral point of view (top) and medial point of view (bottom). Colors indicate the body part that was estimated as the pRF center. The ROIs are denoted by the lines drawn on the surfaces: primary motor cortex (M1), primary somatosensory cortex (S1), supplementary motor area (SMA), dorsal premotor cortex (PMd), ventral premotor cortex (PMv), Insula/Sylvian fissure (Insula), inferior parital cortex (iPC), and superior parietal cortex (sPC).
(TIF)

**S3 Fig. Conventional pRF size maps.** The conventional pRF size is shown on the average subject pial surface (left) and inflated surface (right) from a lateral point of view (top) and medial point of view (bottom). Colors indicate the pRF size. The ROIs are denoted by the lines drawn on the surfaces: primary motor cortex (M1), primary somatosensory cortex (S1),

supplementary motor area (SMA), dorsal premotor cortex (PMd), ventral premotor cortex (PMv), Insula/Sylvian fissure (Insula), inferior parital cortex (iPC), and superior parietal cortex (sPC).
(TIF)

**S4 Fig. Individual non-rigid pRF center maps.** For each participant the non-rigid pRF centers projected on a flattened cortex reconstruction are shown with different colors representing different body parts. The ROIs are denoted by the white lines and text on top of the maps.
(TIF)

**S5 Fig. Individual non-rigid pRF size maps.** For each participant the non-rigid pRF sizes projected on a flattened cortex reconstruction are shown. Darker hues represent smaller pRF sizes and lighter hues represent larger pRF sizes. The ROIs are denoted by the white lines and text on top of the maps.
(TIF)

**S6 Fig. Non-rigid versus conventional pRF sizes.** Comparison between non-rigid (blue) and conventional (pink) pRF sizes across ROIs. The conventional pRF model returns smaller pRF size estimates for areas M1 and S1, while the non-rigid pRF model returns smaller pRF size estimates for the other cortical areas. The error bars denote the S.E.M. across subjects.
(TIF)

**S7 Fig. Relationship pRF size and Connectivity (degree).** Each dot shows the relationship of pRF size (horizontal axis) with the Connectivity metric (vertical axis) of all 18 body part representations (individual dots) for each subject. The different colors correspond to the 8 subjects, for which the linear regression is shown by lines of the same color. Mean Pearson R across subjects for each ROI is presented in the top right corner of each plot.
(TIF)

**S8 Fig. Task design bias.** The normalized distances (1 = pRF center) for all cued body parts (horizontal axes) and all included surface vertices are shown, averaged over the estimated pRF center for each body part (separate panels). The estimated center body part is presented by the orange bar and by default is closest to the pRF center (i.e. highest normalized proximity value). The body parts that were presented during the same run as the estimated pRF center are represented by the green bars, while the body parts presented in the different run as the pRF center body part are shown by the purple bars.
(TIF)

**S9 Fig. Equi-representational graph theory metrics.** Whole-body-graphs are presented per ROI (from left to right) and for the connectivity, clustering and betweenness centrality coefficients (from top to bottom). The colors of each node in the graphs correspond to a specific body part given by the schematic at the far right. The number of surface vertices that contributed to each body part node was held equal. The connections between any 2 body part nodes was calculated per ROI and shown here through the lines connecting the nodes. The thicker the line the stronger the connection between body parts. (A) Connectivity values per body part node and ROI are depicted. The size of the body part nodes presents the size of the connectivity value per node. (B) Clustering coefficients have the same layout and graphs as the connectivity values. Here the size of the body part node reflects the strength of the clustering coefficient. (C) The size of the body part nodes in the ROI graphs reflects the strength of the betweenness centrality coefficient.
(TIF)

**S10 Fig. Equi-representational body part modules.** For each ROI, different modules are represented by different colors. Each module was determined on the basis of an equal number of surface vertices that contributed to the body part nodes. Note that the colors only define a cluster of nodes within one graph, and any correspondence of colors between graphs is purely accidental. The whole-body graph layout is presented at the outmost right indicating the node-body part relationship.
(TIF)

**S1 Text. Task design bias testing.**
(DOCX)

**S2 Text. Equi-representational graph theory methods.**
(DOCX)

## Acknowledgments

We like to thank Mariana Pedroso Branco and Janne Luppi, who helped with data acquisition.

## Author Contributions

**Conceptualization:** Wouter Schellekens, Nick F. Ramsey, Natalia Petridou.

**Data curation:** Wouter Schellekens.

**Formal analysis:** Wouter Schellekens, Carlijn Bakker.

**Funding acquisition:** Nick F. Ramsey, Natalia Petridou.

**Investigation:** Wouter Schellekens.

**Methodology:** Wouter Schellekens.

**Project administration:** Wouter Schellekens.

**Resources:** Wouter Schellekens.

**Software:** Wouter Schellekens, Carlijn Bakker.

**Supervision:** Wouter Schellekens, Nick F. Ramsey, Natalia Petridou.

**Visualization:** Wouter Schellekens.

**Writing – original draft:** Wouter Schellekens.

**Writing – review & editing:** Wouter Schellekens.

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
