## [Decision Letter · Decision Letter 0]

12 Feb 2021

Dear Dr. Schellekens,

Thank you very much for submitting your manuscript "Moving in on human motor cortex. Characterizing the relationship between body parts with non-rigid population Response Fields." for consideration at PLOS Computational Biology.

As with all papers reviewed by the journal, your manuscript was reviewed by members of the editorial board and by several independent reviewers. In light of the reviews (below this email), we would like to invite the resubmission of a significantly-revised version that takes into account the reviewers' comments.

Reviewers generally found the idea interesting and the paper well written. However, all reviewers raised important issues that need to be addressed before the paper can be accepted for publication. These comments demands some additional analysis and clarification of several details of the graph theory based approach for the method to be more convincing. In addition to these comments, please notice that PLOS CB now requires data to be available for download unless there is reasonable restriction (such as sensitive data): https://journals.plos.org/ploscompbiol/s/data-availability Unless the data meet certain criterion (such as sensitive data), they should be made available without restriction instead of upon request.

We look forward to reading the revision of your updated manuscript.

We cannot make any decision about publication until we have seen the revised manuscript and your response to the reviewers' comments. Your revised manuscript is also likely to be sent to reviewers for further evaluation.

Sincerely,

Ming Bo Cai

Associate Editor

PLOS Computational Biology

Daniele Marinazzo

Deputy Editor

PLOS Computational Biology

We are glad to have received your manuscript to PLOS Computational Biology. Reviewers generally found the idea interesting and the paper well written. However, all reviewers raised important issues that need to be addressed before the paper can be accepted for publication. These comments demands some additional analysis and clarification of several details of the graph theory based approach for the method to be more convincing. In addition to these comments, please notice that PLOS CB now requires data to be available for download unless there is reasonable restriction (such as sensitive data): https://journals.plos.org/ploscompbiol/s/data-availability Unless the data meet certain criterion (such as sensitive data), they should be made available without restriction instead of upon request.

We look forward to reading the revision of your updated manuscript.

Reviewer's Responses to Questions

**Comments to the Authors:**

Reviewer #1: This paper makes two novel methodological contributions. 1) They use a 'non-rigid' model to estimate a motor control population response field (pRF) in single fMRI voxels. 2) They apply graph theoretic measures to analyze the relationships between the pRFs in different brain regions. The non-rigid pRF method is an interesting proposal that could have applications beyond the study of motor control. As for the empirical results, my expertise is in modeling the visual system, so my judgment of their novelty is largely based on the author's descriptions and citations. These seem to indicate that the results are in agreement with prior work, with some interesting results but no big surprises or major theoretical advances.

I believe this paper would be suitable to publish at PLOS Computational Biology with major revisions.

The first major point of revision is in interpreting the graph theoretic measures. In the abstract and the introduction, the authors assign each formally defined measure an interpretative term (i.e. connectivity = 'uniqueness', clustering = 'clique-formation', and betweenness centrality = 'importance'). I think this is a useful way to convey the results. However, I think the authors should reconsider or reframe these terms for clustering and betweenness centrality. The weighted clustering coefficient only examines cliques in triangles around some node, which means that it cannot capture any higher-order cliques. Higher-order cliques may actually capture some interesting behavior in motor control regions. One reference about higher-order cliques can be found here: https://www.cs.cornell.edu/~arb/papers/higher-order-clustering-PRE-2018.pdf The authors should either address this limitation in the interpretation, or explore the results with some higher order clustering coefficients.

Additionally I think describing betweenness centrality as 'importance' can be misleading, or at least raises many questions. Important for what? More important for the brain to represent in high fidelity? Important for overall motor control? etc. I accept the author's extended definition in the introduction, as "the (indirect) involvement of a particular body part when other body part moves" (lines 103-104). I think another appropriate description would be that the betweenness centrality measure describes how much the representation of the motion of a single body part overlaps with the motion of other body parts. My best suggestion is to replace 'importance' with something like 'representational centrality' or 'representational overlap'. Either way, descriptions of this measure throughout the Discussion should be revised with this point in mind.

The second major point is a possible error with the manuscript or data. In Figure 6, the authors show graphs for each ROI and graph theoretic measure, where the node sizes correspond to the value of each measure. From my understanding, the edges of the graph should simply be derived from the correlation matrix of the average response fields, which is done separately for each ROI. This implies that the graph edges should be identical in the 3 graphs of each column of Figure 6, but that is not the case. For example, in 6A S1 there is an edge from the abdomen to the knee, but 6B S1 does not have this edge. Why is this the case?

Other miscellaneous points:

- The authors find evidence of somatotopy in all ROIs. However, Fig. 2 suggests that not all regions contain complete representations of the 18 areas (e.g. PMv appears to missing red ankle/toe activation). It would be useful to investigate this, or at least show a more complete plot of the evidence for somatotopy. For each ROI, the authors could plot the x/y coordinates of each voxel (each point colored with the body part preference) with the somatotopy regression line plotted on top.

- Is there some systematic relationship between pRF size and any of the graph theoretic measures? This might be useful to further illustrate what the various graph theory measures add beyond the traditional pRF approach.

- Is there any substantial intersubject variation (particularly for the results in Fig 6)? Along the same lines, it would be nice to see individual points for each subject in Fig 5 instead of / in addition to the SEM error bars.

- Is it possible that the results for voxels preferring some part of the head (body parts 13-18) are affected by a confound with head motion and the fMRI data? Even after motion correction, large movements can cause biases in the fMRI data. How much motion do subjects typically exhibit, and were there outliers to exclude?

- While I do not believe it will change the conclusions of this report, it would be nice to see some examination of the distribution and sphericity assumptions of the ANOVA data. Some sphericity correction may be needed.

Reviewer #2: please, see attachment

Reviewer #3: PCOMPBIOL-D-20-02093

Moving in on human motor cortex. Characterizing the relationship between body parts with non-rigid population Response Fields.

Summary

This is a nicely presented manuscript summarising data from an 7T fMRI experiment and a novel approach to analysing the results. The aim of the study / approach is to shed light on the sensori-motor representations in different areas of human cortex. The method is a modification of the popular population receptive field method, motivated by a graph-theory approach. It manages to capture the fact that maps of sensory / motor response fields (if and when they exist) are don't have a known metric, but are likely to follow a consistent ordering. The non-rigid method presented here allows to estimate the weighted contribution from motion in different body region, regardless of the uneven (and unknown) mapping in sensory space

I have a couple of smaller comments and two questions which might require a little bit of additional thought, but I think overall, this is a very interesting paper that might be of interest to a wider audience.

General comment:

The way the model is presented appeals and I was intrigued by the graph theory analysis that was used to interpret and summarise the pRF fitting findings. On the data fitting / optimisation itself, I think the authors need to explain in a bit more detail how the allowing the different body area / cortical map territories to contribute in this non-rigid model fit is different from simply allowing a separate scale factor for each... thinking through this part of the analysis in detail: different body part responses can contribute to the predicted response in a weighted, rank-ordered way - if my understanding of this is correcrt, then the fitting should work equally well with a profile of y=1-x, x=[0..1], just the estimated d_i parameters would be different, modulo standard gaussian quantiles.... I am not saying this because I think the authors should do it the other way, but suggesting that maybe an additional diagram of a sample model profile showing the non-rigid placement of an example fit solution would help readers follow this logic a bit better. Additionally, knowing how often the d_i parameters bunch up / collapse on a small region of the parameter range would be important to know, as these numbers will be obscured by relying on the gaussian profile and only showing a summary version of the non-rigid part of the fit parameters.

The regions of interest for the network analysis seem a bit ad-hoc (as the authors acknowledge). I understand the motivation for carving up the cortex into these sub-regions that appear to have a somato/mototopic map... but given my comments on appearance of map-like structure (from color maps) below and also the fact that other parcellations are readily available: should going with other parcellations (say, by standard anatomical criteria) not produce similar graph summaries? Or are the results presented here, dependent on / tuned to the choice of ROIs? Maybe explaining this in a bit more detail would help (it's on of the few places where I felt I had to go between results and methods section back and forth a couple of times to understand... so worth smoothing over this).

Specific comments:

I like the approach of showing the full maps of the estimated pRF centers and sized (unthresholded by a correlation or r^2 value). See Figures 2, 3... in particular, this allows everyone to see what the coverage of the fMRI slab was. This therefore avoids the common problem of "no response/value" hiding the fact that no data was acquired for those locations, rather than not passing some statistical criterion. However the flipside of this is that areas of cortex that are unlikely to be tuned to any aspects of sensorimotor function get assigned a value (for centers, ostensibly related to ~hand function... for pRF sizes, some middlish value). For me, two questions spring from this

1. there is clearly some regression / blurring of those estimates towards the mean of the parameter range. Is this an artefact of the optimisation (starting values?) or something else. One way to shed light on this would be to show the goodness of fit / r^2 values for everywhere... which presumably highlights S1/M1, S2, SMA and other areas -- I suggest the authors add at least a panel indicating a statistic like this in the main MS (+ more detailed info in the Supplementary Materials)

2. In both cases, the choice of colormap could probably be different. I am saying this as _some_ maps like jet() or hsv() / or similar maps in matplotlib are not perceptually equivalent at different points on the scale ... the effect is to create perceptual groupings in the displayed maps that don't reflect either the (quasi)-linear scale imposed by the categories (individual sites... or even face... to lower limbs). I think a more natural choice would have been something that has perceptually matched step for each categorical jump - or to go for a colormap that is made for increasing / decreasing scales. An additional detail here is that for many people, a mapping for hot colors to low values and blueish colors to high values (as in Figure 3) is probably the opposite of what they expect. For the size map 1...14, an increasing color map without categorical jumps seems definitely better.

**Have all data underlying the figures and results presented in the manuscript been provided?**

Reviewer #1: **No: **I do not see instructions about where or how to obtain the data, other than "All data can be made available upon request" in one of the submission fields.

Reviewer #2: Yes

Reviewer #3: **No: **Don't see a probelm, just not sure here about policy: available from authors, but not currently in a repo.

PLOS authors have the option to publish the peer review history of their article (what does this mean?). If published, this will include your full peer review and any attached files.

Reviewer #1: No

Reviewer #2: No

Reviewer #3: No
---

## [Decision Letter · Decision Letter 1]

19 Aug 2021

Dear Dr. Schellekens,

Thank you very much for submitting your manuscript "Moving in on human motor cortex. Characterizing the relationship between body parts with non-rigid population Response Fields." for consideration at PLOS Computational Biology.

As with all papers reviewed by the journal, your manuscript was reviewed by members of the editorial board and by several independent reviewers. In light of the reviews (below this email), we would like to invite the resubmission of a significantly-revised version that takes into account the reviewers' comments.

Please address all issues raised by Reviewer 2. Please note that Reviewer 1 pointed out that the code is not shared and made additional suggestion for figure.

We cannot make any decision about publication until we have seen the revised manuscript and your response to the reviewers' comments. Your revised manuscript is also likely to be sent to reviewers for further evaluation.

Sincerely,

Ming Bo Cai

Associate Editor

PLOS Computational Biology

Daniele Marinazzo

Deputy Editor

PLOS Computational Biology

Please address all issues raised by Reviewer 2. Please note that Reviewer 1 pointed out that the code is not shared and made additional suggestion for figure.

Reviewer's Responses to Questions

**Comments to the Authors:**

Reviewer #1: Thank you to the authors for a thorough reply. My major concerns have been addressed.

I would suggest that the authors include the plots of connectivity vs. pRF size in the supplement. I appreciated their discussion in their reply, and believe that the finding would be of interest to some readers. The positive correlation suggests that the author's definition of pRF size, i.e. representational spread across body parts, can predict connectivity, i.e. how related one pRF center is to another.

I think the current version of Figure 6 is acceptable. My suggestion was to actually mark each subject in a different color point in the plot (rather than all points being one color), but I see that it adds enough visual noise that it may not be useful.

Reviewer #2: see attached document

Reviewer #3: The authors have made great and very thorough efforts to address my comments (and I saw they took equal care with for the comments raised by the other two reviewers). The revised manuscripts addresses the points I have raised - I have no further concerns.

**Have the authors made all data and (if applicable) computational code underlying the findings in their manuscript fully available?**

Reviewer #1: **No: **The authors have stated that they are not able to publicly release the data because they do not have consent from the subjects.

I have not seen any code. The authors should be able to release the code used to actually form the graphs, compute the various metrics, and run the statistical tests.

Reviewer #2: None

Reviewer #3: Yes

PLOS authors have the option to publish the peer review history of their article (what does this mean?). If published, this will include your full peer review and any attached files.

Reviewer #1: No

Reviewer #2: No

Reviewer #3: No
---

## [Decision Letter · Decision Letter 2]

15 Nov 2021

Dear Dr. Schellekens,

Thank you very much for submitting your manuscript "Moving in on human motor cortex. Characterizing the relationship between body parts with non-rigid population Response Fields." for consideration at PLOS Computational Biology.

As with all papers reviewed by the journal, your manuscript was reviewed by members of the editorial board and by several independent reviewers. In light of the reviews (below this email), we would like to invite the resubmission of a significantly-revised version that takes into account the reviewers' comments.

We appreciate the extensive effort in addressing the reviewers' comments. The reviewers acknowledge the novelty of the methods proposed in this paper. However, currently one reviewer still has major concerns with respect to the interpretation of the data. We do think that the reviewer made legitimate point, but they were not adequately addressed. If these comments are not taken seriously in the next round of revision, we cannot guarantee accepting the paper.

We agree with the reviewer that the limitation in design that different body parts are tested in different runs cannot be neglected and accounted for post hoc. Therefore, this limitation should be clearly stated in discussion, in addition to the analysis you have provided (demonstrating no significant difference between body part differences presented within the same run and across different runs). We feel that the currently updated sentence still has not sufficiently acknowledged this limitation.

It seems that the suggestion of analyzing each region with equal number of voxels is not a challenging task and may strengthen the paper. You may consider including it to the supplementary material.

Please also take into account other feedbacks from the reviewers.

We cannot make any decision about publication until we have seen the revised manuscript and your response to the reviewers' comments. Your revised manuscript is also likely to be sent to reviewers for further evaluation.

Sincerely,

Ming Bo Cai

Associate Editor

PLOS Computational Biology

Daniele Marinazzo

Deputy Editor

PLOS Computational Biology

We appreciate the extensive effort in addressing the reviewers' comments. The reviewers acknowledge the novelty of the methods proposed in this paper. However, currently one reviewer still has major concerns with respect to the interpretation of the data. We do think that the reviewer made legitimate point, but they were not adequately addressed. If these comments are not taken seriously in the next round of revision, we cannot guarantee accepting the paper.

We agree with the reviewer that the limitation in design that different body parts are tested in different runs cannot be neglected and accounted for post hoc. Therefore, this limitation should be clearly stated in discussion, in addition to the analysis you have provided (demonstrating no significant difference between body part differences presented within the same run and across different runs). We feel that the currently updated sentence still has not sufficiently acknowledged this limitation.

It seems that the suggestion of analyzing each region with equal number of voxels is not a challenging task and may strengthen the paper. You may consider including it to the supplementary material.

Please also take into account other feedbacks from the reviewers.

Reviewer's Responses to Questions

**Comments to the Authors:**

Reviewer #1: Thank you for uploading some version of the code. I am changing my recommendation to Accept since it now meets the journal requirements.

I would like to draw the authors' attention to these recommendations from the journal:

"Authors are encouraged to share their code in a way that follows best practice and facilitates reproducibility and reuse. Authors are encouraged to:

...Share clear documentation alongside the code that details any information needed to run the code, for example dependencies.

State how the code can be accessed in the Data Availability Statement of their manuscript."

Documentation: It seems that the .pro files use Qt. This is not very standard, so some example commands would be useful. In addition, I encourage the authors to document code dependencies, framework version, install requirements, etc. I can already see a couple functions which seem to be dependencies not included or listed anywhere in the repository (e.g. `mpfitfun`).

Access: Please include the repo link somewhere where it will be accessed with the manuscript (e.g. the Data Availability Statement.)

Reviewer #2: I would like to thank the authors for their renewed efforts to address the points I raised.

I answer to some points below:

- Regarding the design bias issues, thank you for making these distance comparisons to evaluate the presence of potential biases. I think it’s a good way to make sure no massive bias was introduced indeed. However, this global measure (across the whole FOV) is unlikely to detect whether these design issues are impacting in more specific ways some cortical regions over others (depending on their function). This cannot be account for a posteriori and hinders interpretation in terms of sensorimotor organisation and relationship between body parts across cortical regions, except if the data is analysed and interpreted separately for the two sets of body parts (i.e., lowerlimb/face on one side and torso/upperlimb on the other).

- Regarding the discrepancies I highlighted relative to previous sensorimotor work, I appreciate the authors’ argument that an active paradigm may yield different estimates and acknowledged that some (e.g., pRF size) are indeed likely to be globally larger (since movements elicit more widespread tactile feedbacks). However, whether an active or passive paradigm is used, the general features of S1 and M1 (i.e., relative differences in cortical magnification, for recent papers see Roux et al, 2018/2020 with intracortical recording or Willoughby et al, 2021 with quantified magnification using fMRI) are preserved. The authors’ argument that “the skin itself is not cued for movement” highlights a misunderstanding of the sensorimotor system. The skin is continuously stimulated and stretched throughout any movement, providing enormous tactile feedbacks, which are crucial to adjust our actions (and increasingly attract interest for Brain-Machine interfaces and robotic devices for that reason). Tactile feedbacks that are processed by S1 in particular, whether inputs are triggered by passive touch or by movement. So, comparison/validation with previous knowledge about S1 organisation is legitimate.

- Regarding face somatotopy, several conflicting results were found in humans since Dreyer et al, 1975, but the most recent work using both imaging and intracortical recordings point towards an upward organisation of the face (Root et al 2021, Roux et al, 2018, 2020). Note that an inverse face organisation is found in monkeys (including macaques as in Dreyer 1975), with instead the lower face neighbouring the hand region (e.g., Manger et al, 2996). So, these cannot be used as a reference for humans (and thus to support present findings).

- Regarding the graph theory issue, I agree with the authors that the same issue applies to other analyses such as RSA, which is why when comparing different parts of the sensorimotor system within participants (i.e., like the comparison between different body part representations), care is taken to select the same amount of voxels/vertices. The point of such analyses is to go beyond simple map descriptions (where magnification, RF size, overlap etc can be described), to investigate the relationship between representations. To do so, potential confounds such as differences in magnification need to be accounted for by selecting the same amount of data, using specific criteria to define them in a similar way across conditions (i.e., x most selective ones or x voxels around the peak/CoG of the representation for instance). Otherwise, we end up mixing several characteristics and cannot compare/interpret the data.

- The editing lines 436-439 (“This finding likely illustrates that the conventional pRF model is forced to widen its Gaussian shape to encompass body parts that are not directly adjacent to each other with respect to the cortical homunculus ordering of body parts (e.g. a hand-mouth coupling in parietal cortex [89]) is misleading. It suggests that conventional pRF models might distort things, while it actually fits better what we know from electrophysiological studies (i.e., increased integration between body parts in higher order areas, as suggested by ref 89 and others, see Iwamura’s work).

**Have the authors made all data and (if applicable) computational code underlying the findings in their manuscript fully available?**

Reviewer #1: Yes

Reviewer #2: None

PLOS authors have the option to publish the peer review history of their article (what does this mean?). If published, this will include your full peer review and any attached files.

Reviewer #1: No

Reviewer #2: No
---

## [Decision Letter · Decision Letter 3]

6 Feb 2022

Dear Dr. Schellekens,

Thank you very much for submitting your manuscript "Moving in on human motor cortex. Characterizing the relationship between body parts with non-rigid population Response Fields." for consideration at PLOS Computational Biology. As with all papers reviewed by the journal, your manuscript was reviewed by members of the editorial board and by several independent reviewers. The reviewers appreciated the attention to an important topic. Based on the reviews, we are likely to accept this manuscript for publication, providing that you modify the manuscript according to the review recommendations.

Sincerely,

Ming Bo Cai

Associate Editor

PLOS Computational Biology

Daniele Marinazzo

Deputy Editor

PLOS Computational Biology

[LINK]

Reviewer's Responses to Questions

**Comments to the Authors:**

Reviewer #2: I would like to thank the authors for the additions made to address the points I raised. In particular for doing the equi-graph analysis. I only have a few remaining requests for clarification and transparency:

- Thank you for adding the paragraph lines 385-395. Please specify that the bias you looked for was assessed across the whole FoV (line 388): e.g., “However, we specifically tested for an unintended task design bias, for which we found no evidence when taking into account the whole field of view”. Idem line 207, e.g., “However, we found no evidence for an artificial coupling of body parts that were cued within a single run across the whole field of view (Welch t(13) = 0.34 p = 0.737).”

Please add the bias estimation analysis and plot presented in the previous review as supplementary material and figure and refer to it in the main text. As explained in my previous comment, the fact that you do not find a consistent bias across the whole FOV does not preclude region-specific biases (that might cancel each other by averaging across the FOV).

- Please provide a quantification of head motion at the beginning of the results section when you say you found more motion for the knee condition line 153 (i.e., mean +/- sd motion for knee vs mean +/- sd motion for other conditions).

- Please mention at the beginning of the results section (either line 150, or in Table 1) that different body parts are tested in the two runs. Idem line 205-206. Make it explicit that different body parts are tested in the separate runs, thank you.

- line 297, please consider Moulton et al, 2009 to add to Dreyer’s reference (for more recent reference in humans).

**Have the authors made all data and (if applicable) computational code underlying the findings in their manuscript fully available?**

Reviewer #2: None

PLOS authors have the option to publish the peer review history of their article (what does this mean?). If published, this will include your full peer review and any attached files.

Reviewer #2: No

Figure Files:

Data Requirements:

Reproducibility:

References:

---

## [Editor Report · Decision Letter 4]

22 Feb 2022

Dear Dr. Schellekens,

We are pleased to inform you that your manuscript 'Moving in on human motor cortex. Characterizing the relationship between body parts with non-rigid population Response Fields.' has been provisionally accepted for publication in PLOS Computational Biology.

Best regards,

Ming Bo Cai

Associate Editor

PLOS Computational Biology

Daniele Marinazzo

Deputy Editor

PLOS Computational Biology

---

## [Editor Report · Acceptance letter]

31 Mar 2022

PCOMPBIOL-D-20-02093R4 

Moving in on human motor cortex. Characterizing the relationship between body parts with non-rigid population Response Fields.

Dear Dr Schellekens,

I am pleased to inform you that your manuscript has been formally accepted for publication in PLOS Computational Biology. Your manuscript is now with our production department and you will be notified of the publication date in due course.

With kind regards,

Anita Estes
